



# Constraints on long-term cliff retreat and intertidal weathering at weak rock coasts using cosmogenic [10]Be, nearshore topography and numerical modelling

Jennifer R. Shadrick[1], Dylan H. Rood[1], Martin D. Hurst[2], Matthew D. Piggott[1], Klaus M. Wilcken[3], Alexander J. Seal[1]

[1]Earth Science and Engineering, Imperial College London, London, SW7 2AZ, United Kingdom
[2]School of Geographical and Earth Sciences, University of Glasgow, Glasgow, G12 8QQ, United Kingdom
[3]Institute for Environmental Research (IER), Australian Nuclear Science and Technology Organization (ANSTO), Lucas Heights, NSW 2234, Australia

*Correspondence to*: Jennifer R. Shadrick (jrs17@ic.ac.uk)

**Abstract.** The white chalk cliffs on the south coast of England are one of the most iconic coastlines in the world. Rock coasts located in a weak lithology, such as chalk, are likely to be most vulnerable to climate change-triggered accelerations in cliff retreat rates. In order to make future forecasts of cliff retreat rates as a response to climate change, we need to look beyond individual erosion events to quantify the long-term trends in cliff retreat rates. Exposure dating of shore platforms using cosmogenic radionuclide analysis and numerical modelling allows us to study past cliff retreat rate across the late-Holocene for these chalk coastlines. Here, we conduct a multi-objective optimisation of a coastal evolution model to both high-precision topographic data and [10]Be concentrations at four chalk rock coast sites to reveal a link between cliff retreat rates and the rate of sea level rise. Furthermore, our results strengthen evidence for a recent acceleration in cliff retreat rates at the chalk cliffs on the south coast of England. Our optimised model results suggest that the relatively rapid historical cliff retreat rates observed at these sites spanning the last 150 years last occurred between 5300 and 6800 years ago when the rate of relative sea level rise was a factor of 5–9 times more rapid than during the recent observable record. However, results for these chalk sites also indicate that current process-based models of rock coast development are overlooking key processes that were not previously identified at sandstone rock coast sites. Interpretation of results suggest that beaches and heterogenous lithology play an important but poorly understood role in the long-term evolution of these chalk rock coast sites. Despite these limitations, our results reveal significant differences in intertidal weathering rates between sandstone and chalk rock coast sites, which helps to inform the long-standing debate of 'wave versus weathering' as the primary control on shore platform development. At the sandstone sites, subaerial weathering has been negligible during the Holocene. In contrast, for the chalk sites, intertidal weathering plays an active role in the long-term development of the shore platform and cliff system. Overall, our results demonstrate how an abstract, process-based model, when optimised with a rigorous optimisation routine, can not only capture long-term trends in transient cliff retreat rates but also distinguish key erosion processes active in millennial-scale rock coast evolution at real-world sites with contrasting rock types.



## 1 Introduction

Rock coasts are dynamic, erosional landscapes that form as a result of landward retreat of bedrock at the coastline (Kennedy et al., 2014) and are often identified by features such as a sea cliff and shore platform. Climate change threatens the stability

of rock coasts and is expected to increase rock coast erosion through accelerations in relative sea level (RSL) rise and increased storminess (Trenhaile, 2011; Oppenheimer et al., 2019). However, the quantification of sea cliff response to climate change is challenging because both marine and terrestrial factors at variable temporal and spatial scales influence rock coast evolution. Improving forecasts of future cliff retreat rates is necessary because of the socioeconomic importance of rock coasts and associated hazards, which are further threatened by climate change and increased urbanisation (Hurst et al., 2016).

Understanding the long-term, antecedent trajectory of rock coast evolution is central to the development of predictive models of cliff retreat that account for a changing climate (Hurst et al., 2016; Trenhaile, 2018).

In order to understand the trajectory of long-term rock coast evolution, the interactions between cliff and shore platform dynamics and their combined impact on cliff retreat need to be considered. Cliff retreat is foremost driven by assailing wave

force at the cliff base, and the cliff-platform junction corresponds to the elevation of maximum horizontal marine erosion (Trenhaile, 2018). The shore platform fronting the sea cliff plays an important role in mediating the wave energy available to erode the cliff. It is, therefore, vital to understand wave and weathering processes working concurrently to result in lowering of the shore platform and landward retreat of the cliff (Sunamura, 1992).

The relative importance of 'waves versus weathering' as the primary control on shore platform development has been debated (Stephenson and Kirk, 1998; Trenhaile and Kanyaya, 2007; Kennedy et al., 2011; Retallack and Roering, 2012; Matsumoto et al., 2018). Where weathering relates to the weakening of the rock material through a combination of physical, chemical and biological processes and wave erosion relates to the physical removal of rock material from the shore platform surface and cliff by means of wave action. The dominance of either wave or weathering processes in rock coast evolution still remains

uncertain, but is recognised to vary in importance in different marine, lithological (i.e., rock type) and climatic settings (Dickson, 2006). In general, weathering processes often dominate in low wave energy, micro-mesotidal, weak lithology and warmer climate rock coast sites, whereas wave processes often dominate in high wave energy, meso- or megatidal, more resistant lithology and temperate climate rock coast sites (Stephenson and Kirk, 2000; Dickson, 2006; Trenhaile and Porter, 2007; Kennedy et al., 2011; Retallack and Roering, 2012).


The erosion of rock coasts is highly contingent on the type and structure of the bedrock present at the coast (Rosser et al., 2013; Sunamura, 2015; Buchanan et al., 2020). In fact, a global database identified rock resistance, rather than climate or marine forcings, as the strongest factor controlling rock coast erosion rates (Prémaillon et al., 2018). Rock coasts located in a weak lithology, such as chalk (Hoek and Brown, 1997), are therefore likely to be most vulnerable to climate change-triggered





accelerations in cliff retreat rates. Correspondingly, Trenhaile (2011) conducted a theoretical investigation into the prediction

of rock coast response to climate change using a numerical model and compared results from different rock resistances. He

concluded that, although absolute cliff retreat rates will be greatest in weaker rock types, the proportional increase in cliff

retreat rates will be greater at rock coasts with more resistant material and historically slower cliff retreat rates (Trenhaile,

2011). Further work is needed to understand the sensitivity of long-term rock coast evolution at real world settings in different

lithologies to external changes in climate.

A methodology that has facilitated the study of site-specific, millennial-scale rock coast evolution is the application of

cosmogenic radionuclide (CRN) analysis on shore platforms. Concentrations of CRNs across a shore platform are proportional

to the time of surface exposure to cosmic rays and rate of erosion, and  long-term cliff retreat rates can be quantified when

coupled with a coastal evolution model (Regard et al., 2012; Hurst et al., 2017). Relatively few studies of CRN applications at

chalk rock coast sites have been conducted to date. However, previous applications of CRN analysis at rock coasts across a

range of rock types were able to date shore platform surfaces (Choi et al., 2012) and quantify long-term cliff retreat rates

(Regard et al., 2012; Rogers et al., 2012; Hurst et al., 2016; Swirad et al., 2020; Duguet et al., 2021; Shadrick et al., 2021). By

quantifying past long-term cliff retreat rates, comparisons can be made to observed historical cliff retreat rates. Using these

techniques, both Hurst et al. (2016) and Duguet (2021) identified modern accelerations in cliff retreat rates for chalk cliffs on

the south coast of England and Normandy coast in France, respectively.

Previous CRN analysis at chalk rock coasts have given novel insights into the long-term cliff retreat, yet these studies have

always assumed a simplistic, steady-state geometric model of coastal evolution. The validity of cliff retreat rates derived from

CRN concentrations are dependent on the coastal evolution models that are used to interpret the measured CRN concentration

data (Trenhaile, 2018). It is important, therefore, to ensure the coastal evolution model that is applied faithfully approximates

cliff and shore platform weathering and erosion processes as accurately as possible across the millennial timescales over which

CRNs accumulate. Furthermore, a rigorous optimisation routine is required to optimise a process-based coastal evolution

model to high-precision [10]Be CRN concentrations and topographic data (Shadrick et al., 2021)


Long-term cliff retreat rates for the past 7000 years, and, in turn, projections of future cliff retreat rates to the year 2100 have

been made for two sandstone, rock coast sites in the UK by combining the best available rock coast morphodynamic model

with simulated CRN accumulation and optimising the model to high-precision CRN concentration measurements and

topographic survey data (Shadrick et al., 2021, in revision). Application of a process-based, transient coastal evolution model

to interpret [10]Be concentrations has revealed a linear positive relationship between cliff retreat rates and the rate of relative sea

level rise (Shadrick et al., 2021). These results indicated that negligible subaerial weathering occurred at these sandstone sites,

which was required for the model to match the measured topography and concentrations at the two sandstone sites (Shadrick

et al., 2021).



Until now, a process-based model has not been used to quantify long-term cliff retreat rates for chalk rock coast sites. Here, we apply a process-based model at contrasting lithological settings, including both chalk and sandstone rock coasts, to provide further insights into the dominant coastal erosion processes acting across millennial timescales at sites with varied rock types, which helps to inform the 'wave versus weathering' debate (e.g. Stephenson and Kirk, 1998; Trenhaile and Kanyaya, 2007; Kennedy et al., 2011; Retallack and Roering, 2012; Matsumoto et al., 2018). This study expands on the application of CRN

exposure analysis of shore platforms in the UK by investigating the long-term rate and nature of cliff retreat for two new UK, chalk rock coast sites: Seven Sisters and St Margaret's. As well as modelling the new datasets, we revisit data from two previously studied UK chalk rock coast sites: Hope Gap and Beachy Head (Hurst et al., 2016). Revisiting previous studies provides a unique opportunity to compare long-term cliff retreat results from dissimilar coastal evolution models. Here, we compare results between different models, as well as cliff retreat rates derived from historical records, which helps to advance

our understanding of how rock coasts evolved both in the long-term past and more recently.

## 2 Background on south coast chalk cliffs

The white chalk cliffs found on the south coast of England are iconic and recognisable landforms in the UK and worldwide. Their towering heights of up to ~165 m at Beachy Head (Robinson, 2020) and white chalk lithology are both visually striking

and extremely hazardous. As a result, the chalk cliffs on the south coast of England are some of the most well-studied rock coast settings in the world (Moses and Robinson, 2011). A range of previous studies located at the south coast chalk cliffs have, for example, quantified: 1) millennial-scale cliff retreat rates using CRN analysis (Hurst et al., 2016); 2) modern cliff retreats rates using historical maps (Dornbusch et al., 2008); 3) modern cliff failure using UAV photogrammetry (Barlow et al., 2017; Gilham et al., 2019); 4) hazard classification and risk assessment (Mortimore et al., 2004b; Stavrou et al., 2011); and

5) platform downwear rates using Micro Erosion Meters (MEM) (Foote et al., 2006).

The retreat of chalk cliffs is primarily a function of episodic cliff collapses, where the type, volume and frequency of collapse is controlled primarily by the chalk type and structure and cliff height (Mortimore et al., 2004b; Dornbusch et al., 2008; Robinson, 2020). The porosity of chalk means that groundwater saturation, rainfall and storms also play an important role in

cliff collapse occurrence at these vulnerable coastlines (Duperret et al., 2004, 2005; Mortimore et al., 2004b). One of the largest recent cliff collapses occurred near Beachy Head in 1999 with a total chalk volume loss of up to 150,000 $m^3$, which produced a debris apron that extended up to 130 m across the shore platform (Mortimore et al., 2004b; Moses and Robinson, 2011; Robinson, 2020). In contrast, smaller collapses of volumes <1000 $m^3$ occur much more frequently and often multiple times within a year (Duperret et al., 2004; Williams et al., 2004; Robinson, 2020). The removal rate of fallen cliff debris is widely

unquantified (Moses and Robinson, 2011). Nevertheless, it is suggested that small-scale falls (<1000 $m^3$) may be removed in a matter of days to weeks, medium-scale falls (1000–10,000 $m^3$) can be removed over a few months, and large-scale falls



(>10,000 m³) may take decades to transport all the material (Mortimore et al., 2004b; Moses and Robinson, 2011). As well as the total volume of fallen cliff debris, the size of debris produced can also influence the rate of removal, but this has not been fully explored (Moses and Robinson, 2011).


The Cretaceous white chalk cliffs and shore platforms contain bands of nodular and continuous sheets of flint parallel to bedding (Robinson, 2020). Because flints are composed of diagenetic silica and are, therefore, more resistant to weathering and erosion than the carbonate chalk itself, beach material is made up of flint gravels left behind after the chalk is eroded as well as sediment from palaeo-sand barriers (Dornbusch et al., 2006a; Mellett and Plater, 2018). These flint beaches, which

have a harder material strength relative to the chalk, are therefore effective abrasion tools that contribute to the erosion of both the shore platform and cliff base (Costa et al., 2006; Robinson, 2020). Conversely, beach material can also act as protection from incoming wave erosion (Trenhaile, 2016; Earlie et al., 2018). There is evidence of beach thinning across these southern English coastlines across the Holocene due to diminishing supplies of flints from cliff erosion (Dornbusch et al., 2006a, 2008), increased regional storminess and the introduction of coastal defences (Hurst et al., 2016).


A review of datasets acquired from a range of techniques, including Micro-Erosion Meter measurements (MEM), laser scanning and historical maps, estimated average cliff retreat rates of 11–87 cm yr$^{-1}$ and average platform downwear rates of 0.8–7.2 mm yr$^{-1}$ for the chalk cliffs and platforms, respectively, at the south coast in England and north in France (Moses and Robinson, 2011). In England, the longest record of historical cliff retreat rates were quantified using digitised cliff positions

from historical Ordinance Survey maps for a time period of 130 years, which calculated an average cliff retreat of 35 cm yr$^{-1}$ for the chalk coastline between Brighton and Eastbourne between 1873 and 2001 (Dornbusch et al., 2008). As well as platform downwearing, step backwearing is another erosional mechanism active on chalk shore platforms. Irregular steps form across the shore platform due to the chalk and flint bedding of variable resistances. The seaward edge of the step erodes landward primarily by means of mechanical wave processes and, although highly spatially variable, the rate of step retreat is of similar

magnitude to platform downwear at the same sites (Dornbusch and Robinson, 2011; Moses and Robinson, 2011; Robinson, 2020).

We focus our study on two new chalk rock coast sites at Seven Sisters and St Margaret's at Cliffe (St Margaret's hereafter) as well as two previously studied chalk rock coast sites at Hope Gap and Beachy Head (Hurst et al., 2016) (Fig. 1). The Seven

Sisters site is located on the south coast in Sussex between the two coastal towns of Seaford and Eastbourne (Fig. 1). This rock coast site is located within the Seaford Chalk Formation within the White Chalk Subgroup, which is composed of weak, fine-grained chalk with extensive bands of nodular and tabular flints (Mortimore et al., 2001) (Fig 1). A series of low-amplitude anticlinal and synclinal structures shape the sinuous cliff forms found at Seven Sisters (Mortimore et al., 2004a). As a result, cliffs range from maximum heights of 60–165 m associated with the termination of valley peaks to minimum heights of 12–

14 m associated with the termination of valley troughs (Robinson, 2020). At this field site, the shore platform extends ~226





m offshore from the cliff base with a beach of ~50 m width overlying the cliff-platform junction. Cliff height reaches ~47 m above the shore platform where we collected samples for CRN analyses. Mean spring tidal range is measured as 5.97 m at the Newhaven tide gauge site (National Tidal and Sea Level Facility, 2021).

The St Margaret's site, on the south-east coast of Kent, is situated ~5 km north along the coastline from Dover (Fig. 1). The rock coast at St Margaret's is located within the Lewes Nodular Chalk formation within the White Chalk Subgroup, which is composed of hard to very hard nodular chalk interbedded with softer chalks and marls (Mortimore, 1987; Mortimore et al., 2001) (Fig. 1). Beach widths range from 15 m to 73 m in the area with the shore platform extending ~195 m offshore from the cliff base along the CRN sampling transect. Just east of the field site is St Margaret's Bay, where substantial beach material

and groyne coastal defences are present. The cliff height is ~60 m directly above the shore platform where we sampled for CRN analysis, but cliff heights reach >90 m ~1 km further west along the coastline. Mean spring tidal range is measured as 5.82 m at the Dover tide gauge site (National Tidal and Sea Level Facility, 2021).




**Figure 1: Locations and regional geology for the St Margaret's and Seven Sisters sites. (A) Location of rock coast sites on the south coast of England and chalk bedrock shown. (B) Regional chalk geology for Kent coastline and St Margaret's site. (C) Regional chalk geology for Sussex coastline and location of new Seven Sisters site and previously studied sites Hope Gap and Beachy Head. (D) Hillshade (LiDAR) draped with aerial imagery (Channel Coastal Observatory, 2021) and [10]Be sample locations at St Margaret's. (E) Hillshade (LiDAR) draped with aerial imagery (Channel Coastal Observatory, 2021) at high tide (platform is submerged) and [10]Be sample locations at Seven Sisters.**


Earth **Surface**
**Dynamics**
Discussions

RSL histories have been supplied from a glacial-isostatic adjustment (GIA) model (Bradley et al., 2011). The three sites on the Sussex coast, including Seven Sisters, Hope Gap and Beachy Head, have the same reference site for the GIA model (Sussex, (Bradley et al., 2011)). The GIA reference site for St Margaret's was Kent (Bradley et al., 2011). All four chalk sites show

very comparable RSL histories and show that RSL was at an elevation ~15 m lower than present day 8000 years BP (Fig. 2). Results from the GIA model for the chalk sites also shows that the rate of RSL rise reached a maximum of ~7 mm yr$^{-1}$ at 8000 years BP; this quickly declined to ~2.6 mm yr$^{-1}$ at 7000 years BP and continued to gradually decline to rates of ~0.3 mm yr$^{-1}$ at present day (Fig. 2).

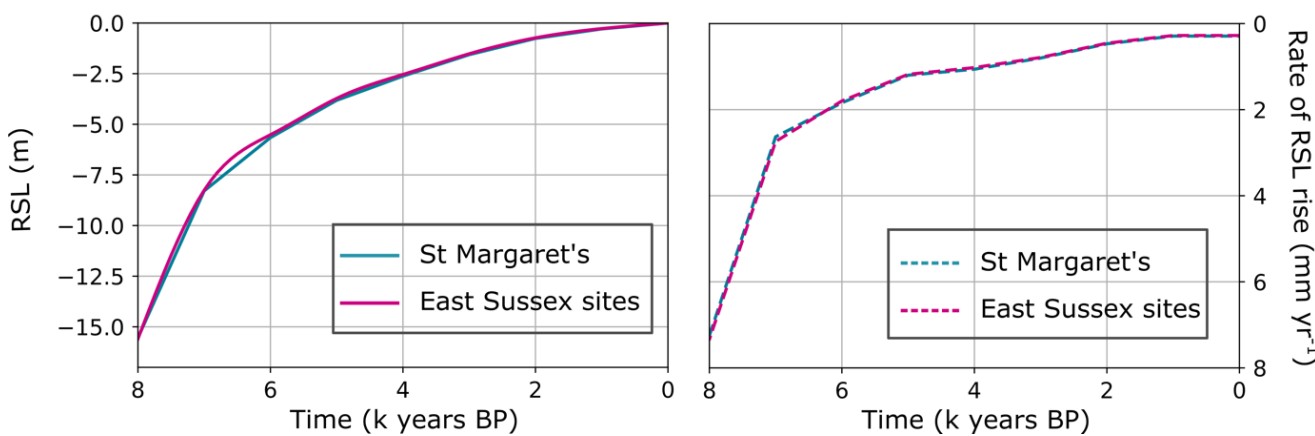


**Figure 2: Relative Sea Level (RSL) (m) and the rate of RSL rise (mm yr$^{-1}$) calculated every 1000 years for the site St Margaret's (blue) and the East Sussex sites, including Seven Sisters, Beachy Head, and Hope Gap (pink). RSL histories were provided by the GIA model of Bradley et al. (2011).**

**3 Methods**

**3.1 Numerical modelling and model optimisation**

Methods associated with our coastal evolution model and multi-objective optimisation approach are provided in detail by Shadrick et al. (2021); however, a basic overview is provided here.

The numerical model couples an exploratory rock coast evolution model (Matsumoto et al., 2016) and a dynamic model for shore platform evolution and $^{10}$Be production (Hurst et al., 2017) to simultaneously model rock coast erosion and $^{10}$Be production. The model applies a dynamic form of coastal evolution that allows for transient cliff retreat, rather than Holocene-averaged cliff retreat rates, in order to quantify a time series of cliff retreat across millennial timescales. Platform morphology is an emergent model element instead of being fixed through model simulation time (e.g. Regard et al., 2012; Hurst et al., 2016; Swirad et al., 2020). Simulated wave erosion is highly simplified by expressing wave hydraulic and mechanical




properties as wave assailing force and follows established conceptual rock coast evolution models (Sunamura, 1992; Trenhaile, 2008; Payo et al., 2015). Erosion of the shore platform and cliff is achieved once wave assailing force exceeds a material resistance value assigned to the rock material. Subaerial weathering is also simulated and works to lower the resistance of the rock material value (Matsumoto et al., 2016). Free parameters chosen to vary within the MCMC simulation were wave

erodibility by means of wave height decay rate ($y$), material resistance ($F_R$) and maximum intertidal weathering rate ($K$). Free parameter selection was informed by previous investigations that found these variables had the greatest influence on whether rock coast evolution was dominated by either wave of weathering-driven erosion (Matsumoto et al., 2018). The defined parameter space was informed by modelling-based and field-based investigations (Sunamura, 1992; Ogawa et al., 2011; Matsumoto et al., 2018; Trenhaile, 2000; Porter et al., 2010a). Altering the value of wave height decay rate ($y$), with units m$^-$

$^1$, varies the spatial distribution and magnitude of wave energy. A greater value for $y$ means wave height will decay more quickly and wave erosion covers a shorter distance across the shore platform, whereas a lower value for $y$ means wave height will decay more slowly and wave energy dissipates across a greater distance across the shore platform. The conceptual value for material resistance ($F_R$), with units kg m$^{-2}$ yr$^{-1}$, simplifies mechanical, geological and structural rock factors into a single value (Matsumoto et al., 2016). Maximum weathering rate ($K$), with units kg m$^{-2}$ yr$^{-1}$, occurs at the mean high water neap tidal

level (MHWN) defined by a weathering efficacy function (Porter et al., 2010a). The range of $K$ explored encompasses a parameter space where negligible intertidal weathering and where weathering rate equal to the material resistance ($F_R$) can be replicated in the MCMC simulations. Maximum weathering rate ($K$) was varied as a function of the material resistance ($F_R$) within the MCMC simulations. The [10]Be concentration is calculated across the shore platform surface and as a function of depth for every annual time step. Both spallation-produced [10]Be at the surface and muon-produced [10]Be at depth are modelled.

Cliff retreat exposes new shore platform material to [10]Be spallation production. Exposure to [10]Be production is modulated through time by the rate of cliff retreat, platform lowering, water cover (including tidal variation and RSL) and topographic shielding (Choi et al., 2012; Regard et al., 2012; Rogers et al., 2012). These factors combine to result in the predicted 'humped' [10]Be concentration profile offshore.

For each site, the two model outputs, including a topographic profile and [10]Be concentration profile, were optimised simultaneously using multi-objective optimisation with the Queso Bayesian calibration library (Estacio-Hiroms et al., 2016) within Dakota optimisation environment (Adams et al., 2019). Before implementing the full optimisation routine, exploration of the parameter space using random sampling was performed to 1) refine the parameter ranges, and 2) tailor the proposal distribution variance for optimal chain mixing with acceptance values of ~23% (Gelman et al., 1997). The proposal distribution

is used to select and move to new samples in the MCMC chain, which directly impacts the acceptance rate. The scaling values, which were used before the topographic and [10]Be concentration residual scores were combined, also impact the acceptance rates. These scaling values are needed to equalize the magnitude ranges of the two single residual scores. The topographic profile scaling value is equal to the standard error from a linear regression of the topographic profile; the [10]Be concentration





scaling value is the average measurement error of the [10]Be concentrations (Shadrick et al., 2021). Table 1 includes inputs and
ranges for values used for the optimisation routine implemented using Dakota for all four chalk sites.

**Table 1: Ranges and proposal distributions (PD) for the free parameters y (wave height decay rate), $F_R$ (material resistance) and K (maximum weathering rate) for the St Margaret's, Seven Sisters, Hope Gap and Beachy Head sites. Maximum weathering rate is varied as a function of material resistance following Matsumoto et al. (2018) (Maximum weathering rate = K x $F_R$). Scaling values for the weighted multi-objective MCMC routine for the topographic profile and [10]Be concentration profile are also shown.**

| Sites | Free parameters | | | | | | Scaling values | |
| --- | --- | --- | --- | --- | --- | --- | --- | --- |
| | $y$ $(m^{-1})$ | | $F_R$ $(kg\ m^{-2}\ yr^{-1})$ | | $K$ $(kg\ m^{-2}\ yr^{-1})$ | | Topographic profile | [10]Be profile |
| | Range | PD | Range | PD | Range | PD | | |
| St Margaret's | 0.01–0.16 | 0.2 | 10–1000 | 0.2 | $1\times10^{-5}$–1 | 0.2 | 0.47 | 877 |
| Seven Sisters | 0.01–1 | 0.3 | 10-1000 | 0.3 | $1\times10^{-3}$–1 | 0.3 | 0.69 | 230 |
| Hope Gap | 0.01–0.16 | 0.1 | 10-1000 | 0.1 | $1\times10^{-5}$–1 | 0.1 | 0.36 | 403 |
| Beachy Head | 0.01–0.16 | 1.2 | 10-1000 | 1.2 | $1\times10^{-5}$–1 | 2 | 0.52 | 855 |

A 10,000 iteration, Metropolis-Hastings MCMC (Metropolis et al., 1953) simulation was used to target a set of model input parameters that produce a model output that best match the measured data. Optimisation is achieved by minimising the negative log-likelihood score from an equally weighted objective function that combines both topographic profile residuals and [10]Be concentration residuals between modelled and measured results. Shadrick et al. (2021) provides a full explanation of how this objective function is formulated and applied within the Dakota environment.

Long-term cliff retreat rates are quantified by taking the best set of input parameters from the MCMC simulation and inputting them back into the RPM model. Uncertainty on best fit results were defined by the 16% and 84% confidence intervals of likelihood weighted posterior distributions of accepted sample positions. Time stamps of modelled cliff positions were back calculated and used to estimate at what time cliff positions occurred at different distances relative to the present-day cliff position.





### 3.2 GIS-based methods for topographic data acquisition

To quantify recent cliff retreat rates for the past ~130 years, georectified historical Ordinance Survey maps were used to digitally map past cliff positions, and then these past cliff positions were compared to the most recent cliff positions taken from aerial photographs. Historical cliff retreat rates were previously quantified between the years 1873 and 2001 across all chalk sites on the south coast (Dornbusch et al. (2008)). We have used these historical cliff retreat rates as a comparison to the long-term cliff retreat rates produced from the coastal evolution model. At St. Margarets, we updated previous historical cliff retreat calculations made by Dornbusch et al. (2008) to include up to the year 2020. Here we used a similar approach to Dornbusch et al. (2008) and used the Digital Shoreline Analysis System (DSAS) 5.0 (Himmelstoss et al., 2018) to quantify cliff retreat rates between the years 1898 and 2020. Historical cliff positions were digitised from OS maps (Digimap, 2021) and present-day cliff positions were digitised from recent aerial photographs and Light Detection and Ranging (LiDAR) imagery (Channel Coastal Observatory, 2021). Historical cliff retreat rates were quantified for ~5.5 km of the coastline that included the site where rock samples for CRN analysis were collected. Using DSAS, a total of 1081 transects were made perpendicular to the shoreline at 5 m intervals to intersect the past and present-day digitised cliff positions. The cliff retreat rate was calculated by dividing the distance between the two cliff lines by the time interval (122 years). The uncertainty is calculated by propagating the uncertainty of the historical and present-day cliff positions in quadrature and dividing by the time interval (Jonah et al., 2016; Himmelstoss et al., 2018).

A modern cross-shore topographic profile of the shore platform was required for comparison to the cross-shore rock coast evolution model (described in section 3.1). The first output of the coastal evolution model that we used (Matsumoto et al., 2016) is a cross-section of the cliff-platform topographic profile with an orientation that is perpendicular to the cliff line. To quantify site specific cliff retreat rates using this model, we need a corresponding topographic profile to which to optimise the model. The Channel Coast Observatory (Channel Coastal Observatory, 2021) provides high quality LiDAR and bathymetry imagery, and so we used this data to construct the topographic cross-section swath profile. From the 1 m spatial resolution LiDAR and bathymetry data, a swath profile of width 10 m and length 300 m was extracted. The location of the swath profile follows the same transect as the [10]Be rock sample collection.

### 3.3 [10]Be sample collection and processing

The second output of the coastal evolution model is a predicted [10]Be concentration profile along the same transect as the topographic profile. Flint nodules, which are composed of amorphous $SiO_2$, in the chalk have the required target elements needed for cosmogenic [10]Be production (e.g., primarily O) (Gosse and Phillips, 2001). Flint samples were collected from the intertidal platforms at Seven Sisters in July 2013 and at St Margaret's in August 2018 (Fig. 3). Before sampling, field





reconnaissance was conducted to locate a flint bed with sufficient number and size of flint nodules that followed roughly the same transect, which extended from the beach toe to the low tide level. We maximised the distance offshore because the offshore sampling distance is proportional to the duration of time to which we can calibrate our coastal evolution model. As exactly as possible, field studies were timed to sample the shore platform at the lowest annual tides to ensure sampling could extend to the furthest offshore and widest width of the shore platform. In-situ, exposed flint nodules were then excavated from the chalk shore platform at ~ 10 m intervals, which resulted in 9 samples at Seven Sisters and 16 samples at St. Margaret's. High-accuracy (1–3 cm) Global Navigation Satellite System (GNSS) coordinates and elevation data were taken at each sample point using a Trimble Geo 7X and external antenna. A laser range finder was used to measure ~10 m intervals between sample points. As well as the shore platform samples, where possible, a sample from a sea cave (Fig. 3b) was also taken to correct for inherited, muogenic-produced $^{10}$Be, i.e., the $^{10}$Be concentration present in the rock before the intertidal platform was exposed by erosion during cliff retreat. We assume the concentration of the sea cave sample at each site is entirely muon-produced and use this as our inheritance correction concentration for the shore platform samples at that site. At Seven Sisters, $^{10}$Be concentrations were corrected for inheritance using the lowest concentration sample closest to the modern cliff because there was not a sea cave (Table 2); this was the same approach taken by Hurst et al. (2016) for the Beachy Head site where a sea cave was not present.





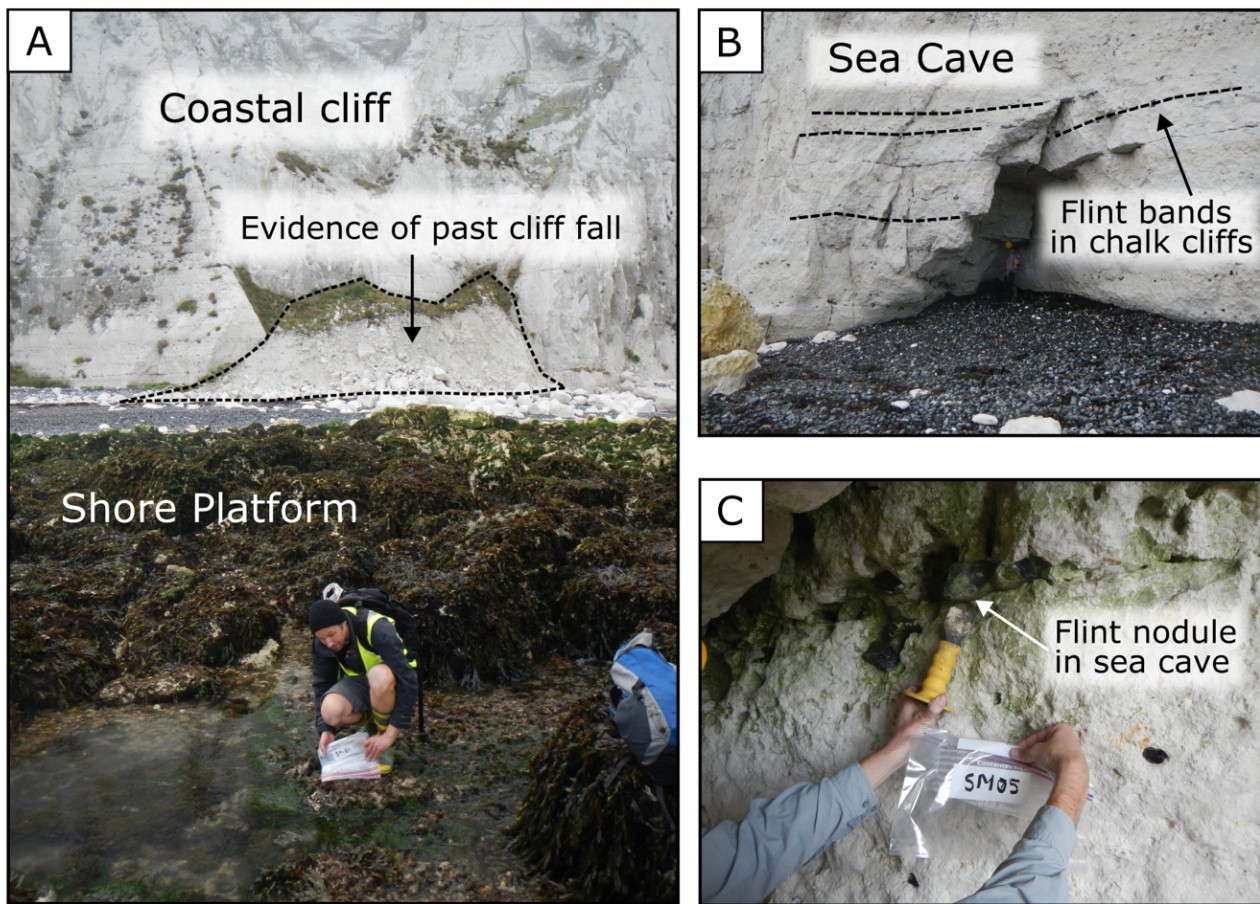

**Figure 3: Photos taken in the field at the St. Margarets site. (A) The chalk coastal cliff and shore platform identified at the site**
**showing evidence of previous cliff fall. (B) Example of flint bands found in the chalk cliffs and sea cave. (C) Photo of sample SM05,**
**which was taken from the sea cave in photo B. Sample SM05 was used as the shielded sample to correct shore platform samples for**
**inherited $^{10}$Be concentrations.**

The shore platform samples were prepared using mineral separation and isotope dilution chemistry methods based on either

standard procedures (Kohl and Nishiizumi, 1992; Corbett et al., 2016) or methods previously used for flint samples (Hurst et

al., 2016) at the CosmIC laboratory at Imperial College London (St Margaret's) and the Scottish Universities Environmental

Research Centre (SUERC) (Seven Sisters), respectively. The $^{10}$Be/$^9$Be analyses by accelerator mass spectrometry (AMS) of

the St Margaret's samples were conducted at the Centre for Acceleration Science at the Australian Nuclear Science and

Technology Organization (ANSTO) using the 6 MV Sirius tandem accelerator (Wilcken et al., 2017). For the Seven Sisters

samples, $^{10}$Be/$^9$Be analyses were conducted at Lawrence Livermore National Laboratory (LLNL) (Rood et al., 2010). For St

Margaret's samples, measured $^{10}$Be concentrations were normalised to the KN-5-3 standard with an assumed ratio of 6.320 x

$10^{-12}$ ($t_{1/2}$=1.36 Ma, (Nishiizumi et al., 2007)). For Seven Sisters samples, measured $^{10}$Be concentrations were normalised to


the 07KNSTD3110 standard with an assumed ratio of 2.85 x 10[-12]. Measured ratios were then corrected for background and inherited $^{10}$Be concentrations using the process blank samples and 'shielded' cliff (St Margaret's) or shore platform (Seven

Sisters) sample, with errors propagated in quadrature, allowing for calculation of absolute $^{10}$Be concentrations (Table 2; Table 3; Table S1; Table S2). Chemistry background blanks for St Margaret's contained 19100–71710 $^{10}$Be atoms, equivalent to 4–40% of total atoms in shore platform samples. For Seven Sisters, chemistry background blanks contained 34794–35599 $^{10}$Be atoms, equivalent to 10–20% of total atoms in shore platform samples.

**Table 2: $^{10}$Be sample and concentration data for Seven Sisters.**

| Sample ID | Location (British Nat. Grid) | | Distance from cliff (m) | Background-corrected concentration $^{10}$Be *(x 10$^3$ atoms g$^{-1}$) | ± 1σ AMS Analytical uncertainty (x 10$^3$ atoms g$^{-1}$) | Inheritance corrected $^{10}$Be ** (x 10$^3$ atoms g$^{-1}$) | ± *** (x 10$^3$ atoms g$^{-1}$) |
|---|---|---|---|---|---|---|---|
| | Easting (m) | Northing (m) | | | | | |
| SS01 | 553791 | 96526 | 219 | 5.06 | 0.23 | 2.29 | 0.23 |
| SS02 | 553813 | 96536 | 203 | 4.41 | 0.23 | 1.64 | 0.23 |
| SS03 | 553818 | 96555 | 185 | 4.88 | 0.25 | 2.11 | 0.25 |
| SS04 | 553827 | 96578 | 160 | 4.38 | 0.24 | 1.60 | 0.24 |
| SS05 | 553814 | 96598 | 146 | 3.70 | 0.20 | 9.22 | 0.20 |
| SS06 | 553812 | 96618 | 128 | 5.70 | 0.26 | 2.93 | 0.26 |
| SS07 | 553811 | 96639 | 110 | 3.11 | 0.17 | 3.32 | 0.17 |
| SS08 | 553824 | 96666 | 78 | 3.11 | 0.21 | 3.31 | 0.21 |
| SS09 | 553818 | 96708 | 41 | 2.77 | 0.15 | - | - |

*Normalised to the 07KNSTD3110 standard with an assumed ratio of 2.85 x 10$^{-12}$. Values corrected for chemistry background using average and standard deviation of two full chemistry blanks CFG1416A and CFG1416B (Table S1) processed in each batch with errors in sample and blank propagated in quadrature. **All SS sample were corrected for inheritance with SS09, the sample closest to the cliff base, assuming little accumulation of $^{10}$Be concentrations. ***Error propagated as $\sigma_c =$

$\sqrt{\sigma_a^2 + \sigma_b^2}$ where $\sigma_a$ is the error of the measured concentration, $\sigma_b$ is the error of the measured concentration used for the inheritance correction SS09.




**Table 3: [10]Be sample and concentration data for St Margaret's.**

| Sample ID | Location (British Nat. Grid) | | Distance from cliff (m) | Background-corrected Concentration [10]Be *(x10³ atoms g⁻¹) | ± 1σ AMS Analytical uncertainty (x 10³ atoms g⁻¹) | Inheritance corrected [10]Be ** (x 10³ atoms g⁻¹) | ± *** (x 10³ atoms g⁻¹) |
|---|---|---|---|---|---|---|---|
| | Easting (m) | Northing (m) | | | | | |
| SM01 | 636869 | 144009 | 153 | 10.94 | 0.91 | 9.65 | 0.94 |
| SM02 | 636866 | 143953 | 167 | 7.99 | 0.88 | 6.71 | 0.92 |
| SM03 | 636828 | 143968 | 126 | 5.62 | 0.83 | 4.33 | 0.88 |
| SM04 | 636819 | 143973 | 116 | 7.23 | 0.84 | 5.93 | 0.89 |
| SM05 | 636761 | 144089 | 0 | 1.97 | 0.79 | - | - |
| SM06 | 636779 | 143992 | 73 | 4.02 | 0.79 | 2.73 | 0.84 |
| SM07 | 636782 | 143991 | 77 | 4.71 | 0.81 | 3.41 | 0.86 |
| SM08 | 636789 | 143983 | 85 | 5.84 | 0.81 | 4.55 | 0.86 |
| SM09 | 636802 | 143974 | 99 | 6.26 | 0.82 | 4.97 | 0.87 |
| SM10 | 636812 | 143972 | 110 | 6.53 | 0.83 | 5.24 | 0.88 |
| SM11 | 636823 | 143971 | 120 | 5.44 | 0.82 | 4.15 | 0.87 |
| SM12-1 | 636835 | 143963 | 135 | 4.51 | 0.81 | 3.22 | 0.85 |
| SM12-2 | 636835 | 143963 | 135 | 5.84 | 0.81 | 4.54 | 0.86 |
| SM13 | 636845 | 143965 | 146 | 3.81 | 0.79 | 2.51 | 0.84 |
| SM14 | 636859 | 143952 | 160 | 7.01 | 0.82 | 5.72 | 0.86 |
| SM15 | 636887 | 143949 | 188 | 6.79 | 0.81 | 5.50 | 0.86 |

*Normalised to the KN-5-3 standard with an assumed ratio of 6.320 x 10⁻¹² (t₁/₂=1.36 Ma, (Nishiizumi et al., 2007)). Values corrected for chemistry background using average and standard deviation of two full chemistry blanks BLK101218, BLK1090119 and BLK2090119 (Table S2) processed in each batch with errors in sample and blank propagated in quadrature.

**All SM sample were corrected for inheritance with SM05, a fully shielded cave sample. ***Error propagated as $\sigma_c = \sqrt{\sigma_a^2 + \sigma_b^2}$ where $\sigma_a$ is the error of the measured concentration, $\sigma_b$ is the error of the measured concentration used for the inheritance correction SM05.

## 4 Results

### 4.1 Historical cliff retreat rates

At the St Margaret's site, historical cliff retreat rates that were quantified using DSAS are on average 7 cm yr⁻¹ with an uncertainty of ± 4.3 cm yr⁻¹ across the ~5.5 km of coastline (Fig. 4). The maximum cliff retreat rate was measured as 29 cm yr⁻¹ at St Margaret's, which coincides with locations that have evidence of cliff failure. The historical cliff retreat rates for all three Sussex coast sites, including Seven Sisters, Hope Gap and Beachy Head, was on average rate 31 cm yr⁻¹, according to

Dornbusch et al. (2008) and are an order of magnitude greater than the historical retreat rate at St Margaret's (Table 4).





**Figure 4: Historical cliff retreat rates (change in cliff edge position) over the period 1894–2020 calculated for ~5.5 km coastline including St Margaret's site using DSAS (Himmelstoss et al., 2018).**





Earth **Surface**
Dynamics
Discussions

**Table 4: Measured historical cliff retreat rates for Seven Sisters, Hope Gap and Beachy Head (Dornbusch et al., 2008, 2006b). Also using DSAS (Himmelstoss et al., 2018) for St Margaret's. Uncertainty and variability across the ~5.5km Kent and East Sussex coastline are also shown.**

| Site | Historical cliff retreat rates (cm yr$^{-1}$) | Uncertainty ± (cm yr$^{-1}$) | Variability (cm yr$^{-1}$) |
|---|---|---|---|
| St Margaret's | 7 | 4.3 | 4–29 |
| Seven Sisters | 39 | 4 | 10–80 |
| Hope Gap | 32 | 4 | 10–80 |
| Beachy Head | 22 | 4 | 10–80 |

## 4.2 Long-term cliff retreat rates

Acceptance ratios of the MCMC chains were between 23% and 31%, which ensures chain mixing and that the parameter space was explored within the optimal range (Gelman et al., 1997). Best-fit results of the three free parameters $F_R$, $K$, and $y$ from the 50%–50%, topographic–$^{10}$Be concentration MCMC inversion are shown in Table 5. For the best-fit results for wave height decay rate ($y$), the slowest wave height decay (0.07 m$^{-1}$) is found for the St Margaret's and Beachy Head sites. The slowest wave height decay rates will result in greatest wave erosion being distributed across the longest distance across the shore platform. The faster wave height decay rate (i.e., rapid wave energy dissipation), as fast as 0.01 m$^{-1}$, is found for the Seven Sisters and Hope Gap sites, which results in less wave erosion localised across a shorter distance and closer to where wave breaking is initialised offshore from the cliff, i.e., when wave height >0.8x water depth. For the best-fit results for material resistance ($F_R$), all sites generally show large uncertainty; however, the exception is for the Seven Sisters site, which shows material resistance must be low (16–127 kg m$^{-2}$ yr$^{-1}$) to match the data sets. Results also show that subaerial weathering ($K$) is active in the long-term evolution of all chalk rock coast sites that were studied with the greatest weathering rates (0.73–4.00 kg m$^{-2}$ yr$^{-1}$) calculated at Beachy Head.



**Table 5: Optimised model parameters from a 50%–50% weighted topographic–[10]Be concentration MCMC inversion. Range of best-fit results expressed as 16%–84% confidence intervals calculated from likelihood-weighted accepted parameter posterior distributions.**

| Site | Wave height decay rate ($y$) (m$^{-1}$) | Material resistance ($F_R$) (kg m$^{-2}$ yr$^{-1}$) | Weathering rate ($K$) (kg m$^{-2}$ yr$^{-1}$) |
|---|---|---|---|
| St Margaret's | 0.03–0.07 | 22–411 | 0.12–0.49 |
| Seven Sisters | 0.01–0.03 | 16–127 | 0.41–0.97 |
| Hope Gap | 0.02–0.04 | 23–350 | 0.48–1.82 |
| Beachy Head | 0.02–0.07 | 22–384 | 0.73–4.00 |


Comparisons between the modelled output and measured data is shown using the best-fit results and uncertainty defined by the 50%–50% weighted topographic–[10]Be concentrations MCMC results (Fig. 4). These results are shown for the present-day timestamp when time is equal to 0 kyr before present (BP) and the present-day cliff position is at 0 m. The best-fit topographic profiles at St Margaret's and Beachy Head best fit the measured data furthest offshore from the cliff; however, the model

output is at a lower elevation to the measured topographic profile further inshore and closer to the cliff base. This mismatch suggests that the gradient of the modelled topographic profile is not as steep as the observed shore platform profile. Most notably, the modelled topographic profiles at the Hope Gap and Seven Sisters sites poorly match the section of the shore platform in the upper-intertidal zone; the modelled elevation is considerably lower than the measured topographic profile in this zone. While the model can replicate the general slope of the platform, it has not been able to capture the topographic step

in the shore platform observed at the Hope Gap and Seven Sisters sites. The measured nearshore platform extending from ~0–200 m from the cliff base is at a higher elevation relative to the general slope of the shore platform with a stepped increase from the offshore platform of ~5 m at Seven Sisters and ~2 m at Hope Gap (Fig. 5). The model best fit results (shown by the solid lines) at these two sites are located at the upper elevation of the model uncertainty. These best fit results relative to the uncertainty range suggests the model was unable to move into a parameter space in the MCMC inversion to match these

elevations while trying to simultaneously match the [10]Be concentrations.

The best-fit model results for [10]Be concentration profiles show the general trends in the [10]Be concentration distributions match the measured data. The best match between modelled results and measured [10]Be concentration data is for the Hope Gap site. In contrast, at the Beachy Head site, although the modelled [10]Be concentration profile replicates the average magnitude in

measured [10]Be concentrations, the model has not captured the peak in [10]Be concentrations (Fig. 5). In fact, the peak in modelled [10]Be concentrations at Beachy Head is ~4500 atoms g$^{-1}$ lower than the measured [10]Be concentration peak. The model also underestimates the peak in [10]Be concentrations by ~2500 atoms g$^{-1}$ at St Margaret's. Moreover, at St Margaret's, a localised drop in measured [10]Be concentrations ~120–150 m from the cliff base is not captured in the modelled [10]Be concentrations.





Similar to the topographic results, the $^{10}$Be concentrations closest to the cliff base cannot be well matched, especially at the

Beachy Head and Seven Sisters sites. Specifically, the model is overestimating the $^{10}$Be concentrations by at most ~1000 atoms

g$^{-1}$ at ~50–150 m offshore from the cliff, in the upper-intertidal zone at both Beachy Head and Seven Sisters sites. Also similar

to the topographic results, the best fit results are at the lowest limit of the model uncertainty for $^{10}$Be concentrations. Best-fit

results found at the outer limits of the uncertainty range further suggests that optimisation of the topography and $^{10}$Be

concentrations occurred in contrasting areas of the parameter space, and, therefore, the model was unable to simultaneously

optimise both datasets.

**Figure 5: Best-fit model results from a 50–50% topographic–$^{10}$Be concentration weighted MCMC inversion for chalk sites at St**
**Margaret's, Hope Gap, Beachy Head and Seven Sisters. Dark lines show best-fit results and shaded areas show the confidence**
**interval uncertainty range. The 16%–84% confidence interval for each free parameter in the MCMC inversion (F$_R$, K, and y) was**
**simulated against the median results for the other parameters. The shaded uncertainty was constructed from the upper and lower**
**limits of the model outputs. For both the topographic profile and $^{10}$Be concentrations, the width of the modern-day, 300 m intertidal**
**shore platform is shown. The modern-day cliff position is at 0 m. The panels on the left side compare the modelled results (coloured)**
**to the measured data (black line) of the topographic profile. The panels on the right side compare the modelled results (coloured) to**
**the measured data (black scatter) of the $^{10}$Be concentration profile.**



The long-term trend in these new cliff retreat rates across the past 7000 years reflect the trend in the rate of RSL for all four chalk sites (Fig. 2, Fig. 6). For all chalk sites, cliff retreat rates are fastest at 7000 years BP, then decline most rapidly between 7000 years BP and 6000 years BP and then continue to decline gradually to present-day. Overall, the slowest cliff retreat rates are calculated for the St Margaret's site where cliff retreat rates decline from 15–35 cm yr$^{-1}$ at 7000 years BP to 1–3 cm yr$^{-1}$ at
present-day. The Hope Gap and Beachy Head sites show similar patterns of cliff retreat rates through time, which decline from 20–55 cm yr$^{-1}$ at 7000 years BP to 1–5 cm yr$^{-1}$ at present-day. The fastest cliff retreat rates are calculated for the Seven Sisters site where cliff retreat rates of 60–110 cm yr$^{-1}$ at 7000 years BP were up to approximately seven times faster than the other chalk sites at 7000 years BP. The cliff retreat rates at Seven Sisters follow the same trend and decline to 4–12 cm yr$^{-1}$ at present-day. The greatest decline in cliff retreat rates is also seen at Seven Sisters where cliff retreat rates declined by as much as 27
times the cliff retreat rate from 7000 years BP to present-day.

Time stamps of cliff positions were back calculated to estimate the duration of time required to erode the present-day, ~250 m wide intertidal platform. According to these best-fit MCMC results, the slowest cliff retreat rates at St Margaret's eroded the present-day intertidal platform during the past ~5300 years. Faster cliff retreat rates modelled at Hope Gap and Beachy Head
eroded the present-day intertidal shore platform during the past ~4100 years and ~4000 years, respectively. The fastest cliff retreat rates modelled at Seven Sisters erode the present-day intertidal shore platform during the past ~1800 years. Overall, the low [10]Be concentrations and long-term cliff retreat rates at all four of these chalk sites confirm the observed shore platforms at these sites are Holocene features and not reoccupied from a previous interglacial period.

Earth **Surface**
**Dynamics**
Discussions

EGU



**Figure 6:** Time series of cliff retreat rates (m yr⁻¹) and uncertainty shown by the solid line and shaded area from 7000 years BP to present day. Cliff retreat rates are calculated from modelled cliff positions every 100 years. The time-period between 8000 years BP and 7000 years BP is excluded as this corresponds to the burn-in period of the model. The cliff retreat rates highlighted by the dashed-line box are shown at a larger scale on the right and correspond to the distance across the shore platform over which measured data were analysed (~250 m).





## 5. Discussion

Our results show that long-term cliff retreat rates reflect the rate of RSL rise across the Holocene for all 4 chalk sites. Comparisons between these model results and historical observations further support previous findings of a recent acceleration in cliff retreat rates at the south coast (Hurst et al., 2016). Our results also reveal contrasting results between chalk and previously studied sandstone sites (Shadrick et al., 2021, in revision) regarding the key erosional mechanisms controlling long-term evolution. However, best-fit model results for the chalk sites have inconsistent, and, in some cases, relatively poor fit to the measured data, which contrasts the well-fit model to data comparisons at the sandstone sites (Shadrick et al., 2021). Below, we make comparisons between long-term cliff retreat trends at rock coast sites with differing lithologies using our dynamic model, as well as comparisons between different model results of cliff retreat rates at our same chalk sites. We also discuss what factors are likely to contribute to the contrasting success of the model when applied to different lithologies. Nevertheless, despite these potential limitations, we are still able to use results from our dynamic, process-based model to identify key distinctions in erosion processes active across millennial timescales at coasts with differing lithology.

### 5.1 Long-term trends in cliff retreat rates

Our model results for the chalk sites are consistent with previous results for two sandstone sites (Bideford and Scalby), and suggest that long-term trends in cliff retreat rate are reflective of the rate of RSL rise (Shadrick et al., 2021). As a result, the model suggests that the fastest rates of cliff retreat for the late-Holocene are found when the rates of RSL are greatest. Due to similar RSL histories for all chalk rock coast sites, the greatest rate of RSL rise during the model simulation time is found at 7000 years BP (Fig. 5). We have excluded the time period between 8000- and 7000-years BP as this interval occurred during the model's burn-in period. For the two sandstone sites previously studied, Bideford and Scalby, best-fit cliff retreat rates were ~5.2 and ~14.4 times faster, respectively, at 7000 years BP in comparison to historical rates of cliff retreat (Shadrick et al., 2021). As for the chalk sites, cliff retreat rates were ~1.2 times faster at Hope Gap, ~1.4 times faster at Beachy Head, and ~2.1 times faster at Seven Sisters 7000 years BP in comparison to historical rates of cliff retreat. These results are generally consistent with Trenhaile's (2011) findings, which concluded that a greater proportional increase will be found at historically slower sites in relation to increased rates of RSL rise. However, Trenhaile's (2011) predictions are for future cliff retreat response to projected accelerations in RSL rise, and are entirely theoretical with no calibration to measured data. In contrast, here, we have used empirical data to reconstruct how cliff retreat responded in the past when rates of RSL rise were comparable to these future projections.

### 5.2 Comparisons of long-term to historical cliff retreat rates

Contradictory to the long-term trends in modelled cliff retreat rates that reflect the rate of RSL rise, the discrepancies between long-term modelled and short-term, historical observed cliff retreat rates suggest that RSL does not have the greatest control of cliff retreat rates at these chalk sites. Comparisons between short-term, historical observed cliff retreat rates, long-term retreat rates derived from the dynamic model, and, where available, long-term retreat rates from a geometric model (Hurst et





al., 2016) (see section 5.3) are shown (Table 6). Using the steady-state model, Hurst et al. (2016) identified an order of
magnitude increase in recent cliff retreat rates compared to long-term, Holocene-average rates at Beachy Head and Hope Gap.
This previous work is consistent with our new results from the dynamic model, which suggests that long-term rates of cliff
retreat across the late-Holocene are an order of magnitude less than the historical rates quantified by Dornbusch et al. (2008).
Furthermore, our antecedent cliff retreat rates quantified using the dynamic model reveal that the historical rates of 22 cm yr$^-$
$^1$ at Beachy Head last occurred ~6400 years BP and historical rates of 32 cm yr$^{-1}$ at Hope Gap last occurred ~6800 years BP.
The rate of RSL rise during these times of accelerated cliff retreat were 2.2 mm yr$^{-1}$ at Beachy Head and 2.6 mm yr$^{-1}$ at Hope
Gap, which are approximately 7.3 and 8.7 times faster than the rate of RSL rise experienced during the observational record,
respectively. Similarly, observed rates of 39 cm yr$^{-1}$ at Seven Sisters last occurred ~6200 years ago when the rate of RSL was
2 mm yr$^{-1}$, which is 7 times faster than the rate of RSL rise over the observable record. The acceleration in recent observed
cliff retreat rates must, therefore, be caused by additional factors other than accelerations in RSL rise (see section 5.5).

Although recent observations of cliff retreat rates at St Margaret's are not as fast as those on the Sussex coast, by using the
dynamic model, we calculate these recent rates of cliff retreat (7 cm yr$^{-1}$) were last experienced at St Margaret's 5300 years
BP. At this time, the rate of RSL rise was ~1.4 mm yr$^{-1}$, which is nearly 5 times the rate of RSL rise experienced during the
past 122 years (0.3 mm yr$^{-1}$) over which the historical cliff retreat rates were quantified. The slower historical cliff retreat rates
calculated at St Margaret's in comparison to the Sussex coast sites could be as a result of the harder Lewes Nodular chalk
lithology at St Margaret's, in comparison to the soft, low-density and high-porosity Seaford Chalk Formation on the Sussex
coast (Mortimore et al., 2004a). Furthermore, Dornbusch et al. (2006b) suggests that cliff retreat along the Kent coast is linked
to larger scale erosion events with return periods longer than the observational record. Larger scale erosion events with longer
return periods can, therefore, result in apparent slower retreat rates if a mass erosion event has not occurred within the survey
period. In contrast, episodic, large erosion events can also result in apparent faster retreat rates if a mass erosion event has
occurred within the survey period relative to the long-term trend in cliff retreat, which requires integration across multiple
events to quantify. Return periods that are longer than the observational record again highlights the relatively large uncertainty
in short records and the need for cliff retreat rates quantified across millennial timescales (Sunamura, 2015).





**Table 6: Comparisons between historical cliff retreat rates and long-term cliff retreat rates derived from both a dynamic model and a geometric model. Note the decline in long-term transient cliff retreat rates follows the pattern of RSL rise rate and the long-term steady-state cliff retreat rates are a step change (see Fig. 7).**

| Site | Historical cliff retreat rates | | Long-term, dynamic cliff retreat rates | | Long-term, geometric cliff retreat rates | |
|---|---|---|---|---|---|---|
| | Time period (years BP) | Rate (cm yr$^{-1}$) | Time period (years BP) | Rate (cm yr$^{-1}$) | Time period (years BP) | Rate (cm yr$^{-1}$) |
| St Margaret's | 122 | 7 ± 4.3 | 5300 | 7 to 3 | - | - |
| Seven Sisters | 128 | 39 ± 4 | 1800 | 18 to 13 | - | - |
| Hope Gap | 128 | 32 ± 4 | 4100 | 10 to 2.5 | 4315 | 5.7 to 1.3 |
| Beachy Head | 128 | 22 ± 4 | 4000 | 10 to 4 | 6139 | 2.6 to 30.4 |


## 5.3 Geometric retreat versus dynamic retreat evolution

Long-term cliff retreat rates were previously quantified for the chalk coasts at the Hope Gap and Beachy Head sites using a geometric coastal evolution model (Hurst et al., 2016). This geometric model was optimised to the same $^{10}$Be concentration datasets used here. We next compared the best-fit cliff retreat rates derived for both our new dynamic and the previous

geometric model and discuss their key similarities and differences (Fig. 7).

At Hope Gap, both the geometric and dynamic models reveal that cliff retreat rates declined during the Holocene, but the trend of decline differs between each model (Fig. 7). The best-fit result from the geometric model finds a step-change in cliff retreat rates from 5.7 cm yr$^{-1}$ to 1.3 cm yr$^{-1}$ at 308 years BP (Hurst et al., 2016). The best-fit results from the dynamic model show a

decrease in cliff retreat rates from 10 cm yr$^{-1}$ 4100 years ago to 3 cm yr$^{-1}$ at present day. Although the geometric model cannot capture transient cliff retreat like the dynamic model, both models show the 250 m intertidal platform at Hope Gap was formed over comparable timeframes: ~4100 years using the dynamic model and ~4315 years using the geometric model (Fig. 7). Moreover, the present-day cliff retreat rates found by both models are within uncertainty of each other: 1–2.4 cm yr$^{-1}$ from the steady-state model and 2–4 cm yr$^{-1}$ using the dynamic model. These comparable results found at Hope Gap validate both

models' findings of declining rates of cliff retreat across the late Holocene. Nevertheless, neither model captures the recent acceleration evidenced with historic observations (see section 5.2)



At Beachy Head, results for the two models do not agree as well. The best-fit result from the geometric model finds a step-change in cliff retreat rates with a significant increase in cliff retreat rates from 2.6 cm yr[-1] to 30.4 cm yr[-1] at 293 years ago

(Hurst et al., 2016). In contrast, the best-fit results from the dynamic model shows a declining trend in cliff retreat rates. The dynamic model reveals best-fit cliff retreat rates that fall from 10 cm yr[-1] at ~4000 years ago to 4 cm yr[-1] at present-day. Observed cliff retreat rates for the past 130 years at Beachy Head were previously quantified as 22 cm yr[-1] (Dornbusch et al., 2008; Hurst et al., 2016), which agrees with the geometric model's scenario of a step increase in cliff retreat rates to 30.4 cm yr[-1] at 293 years ago. This step increase in cliff retreat rates had to be forced in the geometric model to match the low $^{10}$Be

concentrations <150 m from the cliff. We, however, cannot force changes in cliff retreat rates in the dynamic model because cliff retreat rates are emergent from the topographic evolution that is controlled by physical erosion processes. Moreover, the best-fit $^{10}$Be concentration results found for the dynamic model at Beachy Head could not match the inshore low concentrations (Fig 5). This mismatch between measured and modelled $^{10}$Be concentrations could support the finding that there was a significant increase in cliff retreat rates in the recent past at Beachy Head that could not be captured by the processes

represented in the dynamic model.

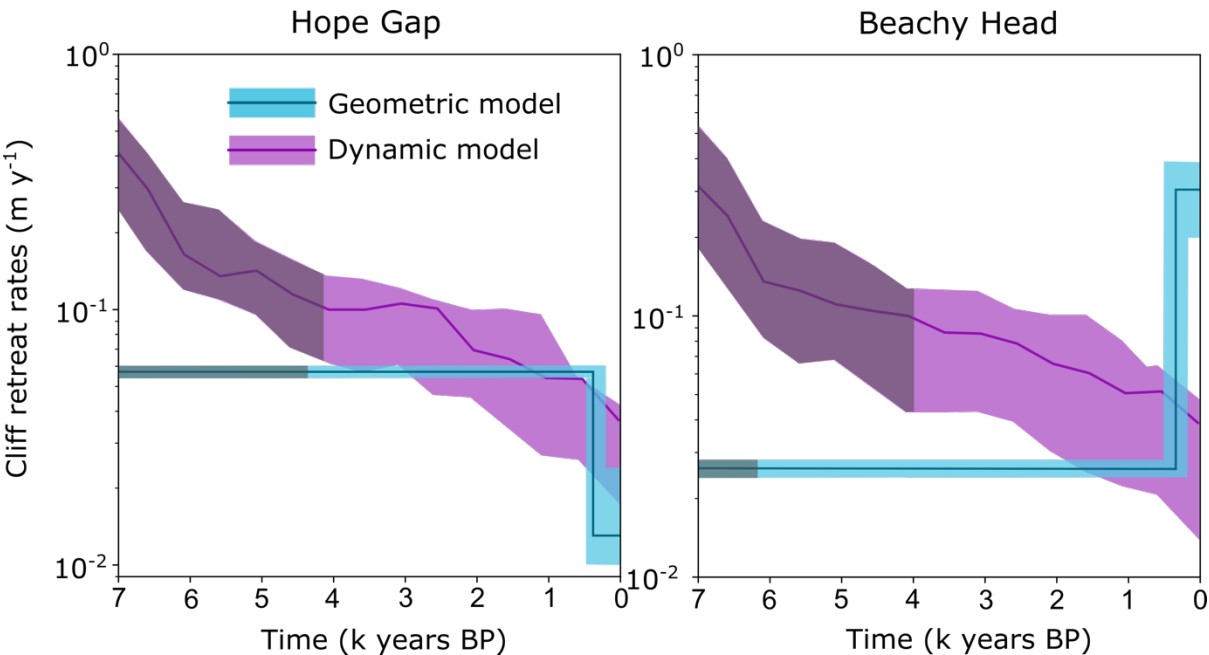

**Figure 7: Comparisons between cliff retreat rates produced from a dynamic coastal evolution (Matsumoto et al., 2016) model and a geometric coastal evolution model (Hurst et al., 2016) for sites Hope Gap and Beachy Head. Cliff retreat rates (m yr⁻¹) are shown**
**for 7000 years BP to present day. The shaded regions highlight the timeframes that extend beyond the measured data in the past.**



### 5.4 Comparisons between erosion processes at chalk and sandstone rock coasts

Unlike the two previously studied sandstone rock coast sites (Bideford and Scalby; Shadrick et al., 2021), all the chalk sites in this study (St Margaret's, Hope Gap, Beachy Head and Seven Sisters) show well-defined normal posterior distributions for weathering rates (Fig. 8). For all four chalk sites over the 8000-year simulation time, we calculated that if the magnitude of

weathering rates fell below approximately $5^{-5}$ x material resistance ($F_R$) (kg m$^{-2}$ yr$^{-1}$) (shown by the dashed line in Figure 8), the material resistance of the rock cells could not be lowered enough by intertidal weathering for waves to erode the rock cells. In other words, when $K<5^{-5}$ as a fraction of material resistance, intertidal weathering has limited influence on reducing rock cell material resistance and wave erosion is the dominant erosion process. In contrast, for the previous sandstone site at Bideford, zero weathering ($K<5^{-5}$) had to occur to match both the topographic and $^{10}$Be concentration dataset (Shadrick et al.,

2021). The $^{10}$Be concentrations at the other previous sandstone site at Scalby could be matched with active weathering, hence the spread of accepted $K$ samples above $5^{-5}$ (Fig. 8). However, like at Bideford, only with negligible weathering could the $^{10}$Be concentrations and topographic data be matched simultaneously at Scalby (Shadrick et al., 2021). For both previous sandstone sites, once weathering rates become negligible ($K<5^{-5}$), there is no change to the model output, which results in broad, near-uniform distributions in $K$ to which the model is insensitive (Fig. 8). For all chalk rock coast sites, however, the best-fit results

contrast with the sandstone sites and show that active subaerial weathering is needed to match the topographic and $^{10}$Be concentration data.

Greater weathering rates at chalk rock coasts compared to sandstone rock coast is supported by field measurements of platform downwear using Micro-Erosion Meters (MEM), laser scanning and Structure-from-Motion (SfM). Although not as commonly

studied as shore platform downwear rates at chalk lithology coasts, downwear rates of harder lithologies tend to have lower rates compared to softer chalk: for example, downwear rates of 0.25 mm yr$^{-1}$ were recorded for a sandstone platform (Yuan et al., 2020); 0.242 mm yr$^{-1}$ for a mudstone and siltstone platform (Porter et al., 2010b); 0.528 mm yr$^{-1}$ for a mudstone, sandstone and shale platform; and 0.625 mm yr$^{-1}$ for a greenschist platform (Mottershead, 1989). This is in contrast to overall higher rates of platform downwear measured at chalk sites; for example, average chalk platform erosion rates of 0.791–7.202 mm yr$^{-1}$

measured across 18 sites (Moses and Robinson (2011)). However, these rates of downwear measured in the field do not exclusively relate to intertidal weathering. In the dynamic coastal evolution model used here, vertical downwear is accounted for by 1) the vertical component of wave assailing force (Stephenson and Kirk, 2000; Trenhaile, 2000; Matsumoto et al., 2016) and 2) intertidal weathering defined by a weathering efficacy function (Porter et al., 2010a; Matsumoto et al., 2016). The down-wearing component of wave erosion follows Stephenson and Kirk (2000) and Trenhaile (2000), and is proportional to

the back-wearing force at the still water level, with intensity declining exponentially with water depth (Matsumoto et al., 2016). The weathering efficacy function (Porter et al., 2010a) dictates that maximum weathering occurs at the mean elevation of the lowest high tide and efficacy decreased above and below this elevation (Matsumoto et al., 2016). The implications of the model's representation of subaerial weathering are further discussed in section 5.4.1.





Furthermore, the best-fit results for material resistance for the new chalk sites show wide distributions (Table 5). This wide

distribution of best-fit material resistance was also the case for the previous sandstone sites (Shadrick et al., 2021) and is caused

by correlation between the free parameters, especially between material resistance and wave height decay rate (Shadrick et al.,

2021). Due to the dynamic model's abstract representation of physical rock qualities and erosion processes, these results show

that material resistance alone is not sufficient to make distinctions between different lithologies at rock coast sites. Clear

distinctions for the weathering rate shown here between the sandstone and chalk sites show that weatherability of the shore

platform at the rock coast site is a more significant indicator than the material resistance when assessing long-term platform

downwear.

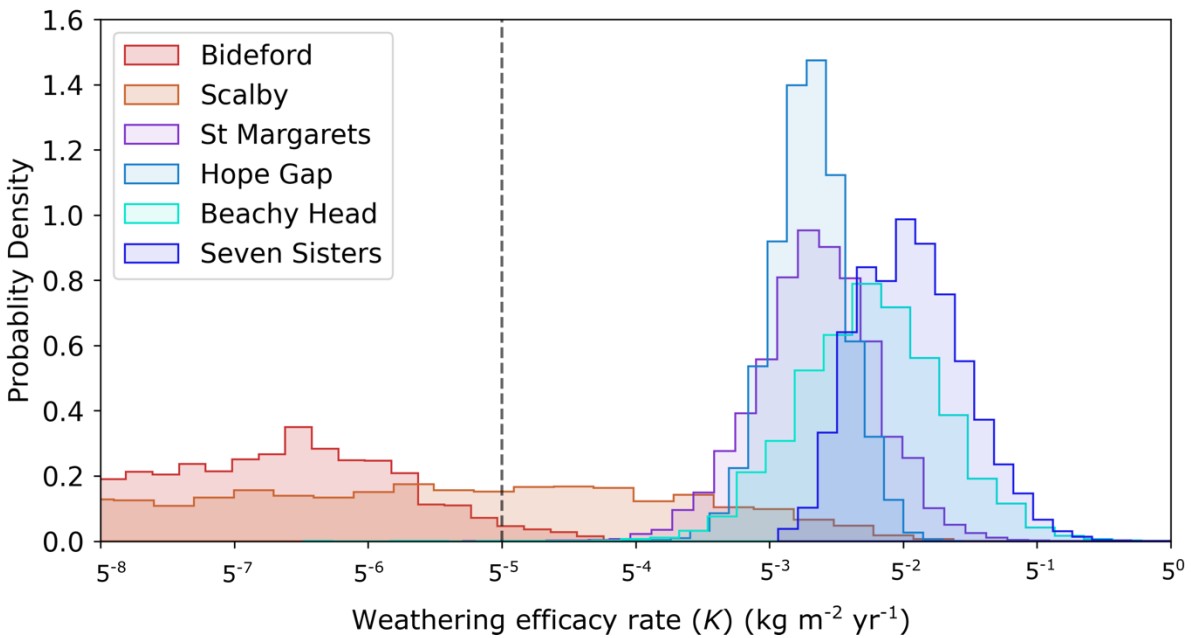

**Figure 8: Posterior histograms of likelihood-weighted, accepted MCMC weathering efficacy rate (K) samples. Here weathering rate is calculated as $F_R$ x K. Red and orange histograms show the results for the two sandstone sites: Bideford and Scalby. Blue, purple, green histograms show the results for the four chalk sites: St. Margarets, Hope Gap, Beachy Head and Seven Sisters. Above the dashed line (where K<5$^{-5}$) is when weathering is active across the shore platform. Below the dashed line (where K<5$^{-5}$) is when weathering processes are negligible.**


## 5.5 The representation of near-shore processes

The best-fit results derived from the dynamic model for the chalk sites, particularly seen at sites Hope Gap, Beachy Head and

Seven Sisters, have been unable to simultaneously match both the topographic and $^{10}$Be datasets. This is in contrast to highly



comparable modelled and measured results for the previously studied sandstone sites (Shadrick et al., 2021). These results suggest that additional processes not yet implemented within the model are important in the long-term evolution of chalk rock coasts, which play a comparatively minor role at the sandstone rock sites previously studied. At the Beachy Head site, we suggest that the misfit between observed and modelled [10]Be concentrations close to the cliff base is caused by an acceleration

in recent cliff retreat rates that could not be captured by the dynamic model (see section 5.3). Similarly, at Seven Sisters, because the dynamic model could also not match the low [10]Be concentrations closest to the cliff (Fig. 5), this mismatch could also suggest there was a recent increase in cliff retreat rates at the Seven Sisters site as well. This explanation, however, may not exclusively account for the over-predicted nearshore [10]Be concentrations calculated by the model, and also may not explain the mismatch in nearshore topography (Fig. 5). Below, we discuss the representation of nearshore processes used in the model,

as well as processes overlooked by the model, that may explain the discrepancies between the modelled results and measured data at the chalk sites.

### 5.5.1 Intertidal weathering and other contributions to platform erosion

Tides control the duration and frequency over which different elevations of the shore platform are immersed and exposed by water. Tidal controlled variations in water level mean the efficacy of weathering in the form of wetting and drying, salt and frost weathering, i.e., processes controlled by cycles of water exposure, that vary with elevation across the shore platform (Stephenson and Kirk, 2000; Trenhaile, 2018). Weathering of the shore platform is exclusively considered by the model as tidally controlled weathering processes, which are represented as a weathering efficacy shape function that dictates maximum

weathering occurs at the lowest high tide level and decreases above and below this elevation (Porter et al., 2010a; Matsumoto et al., 2016, 2018). This weathering function is based on results from laboratory experiments of wetting and drying and salt weathering (Porter et al., 2010a). However, field results do not match this spatial pattern of weathering efficacy; for example, corresponding MEM field measurements (Porter et al. (2010a)) and further comparisons of field-measured downwear rates across various studies (Trenhaile, 2003, 2018) found no significant relationship between platform downwear and elevation in

relation to tides. In the field, platform downwear is also influenced by frost, abrasion, bioerosion and rock surface swelling, which all have variable efficacies in relation to elevation that may explain differences found between the laboratory and field results (Trenhaile, 2018; Porter et al., 2010a). Studies on the south coast of the UK all found greatest downwear on chalk platforms to be at the top of the platform in the zone of beach-supplied abrasion (Ellis, 1986; Andrews, 2001; Foote et al., 2006). The influence of beach material on shore platform erosion is discussed in the following section (see section 5.5.2).


It is suggested that intertidal weathering mainly dictates platform downwear where abrasive material and bioerosion are absent (Trenhaile, 2018). This cannot be said for the chalk shore platforms studied here because there is evidence of abrasive beach material and biological activity. Both abrasion and bioerosion are not included in the coastal evolution model used here. Abrasion of the shore platform surface is especially efficient when beach material is harder than the bedrock it is eroding



(Costa et al., 2006; Robinson, 2020). Because this relative hardness is the case between flint gravel beaches and chalk shore platforms at our south coast sites, abrasion from beach material is likely to contribute to considerable platform downwear and is discussed in the following section (see section 5.5.2). Offshore from the abrasion zone, it is thought that bioerosion, e.g., changing pool chemistry, weakening fractures in rocks, and boring into the platform surface (Naylor et al., 2012; Trenhaile, 2018) may even dominate shore platform erosion (Foote et al., 2006; Henaff et al., 2006; Robinson, 2020). Because bioerosion

is so spatially varied, however, the overall contribution to platform downwear remains uncertain, especially in non-tropical environments and in the subtidal/submarine zone (Trenhaile, 2018). Future research into the distribution of species in the intertidal zone on chalk platforms may help to shed light on bioerosion-controlled downwear at these sites.

Our modelling results and relevant literature suggest that our dynamic model's representation of platform downwear, which is

controlled principally by a single function describing intertidal weathering in the model, may not be the most appropriate mechanism of platform downwear for these chalk coast sites. Nevertheless, clear differences between best-fit modelled weathering rates at chalk and sandstone sites (Fig. 8) reveal that long-term platform downwear occurred at much faster rates at the chalk sites. More work is needed to establish which specific processes are more dominant at the chalk sites in order to result in greater platform downwear.


### 5.5.2 Beach material and cliff debris

Beach material and cliff debris have the potential to both enhance and limit erosion of the shore platform and cliff base (Sunamura, 1982; Limber and Murray, 2011). Evidence of beaches that enhance erosion include field studies that have measured downwear with MEM's on chalk platforms; these field studies have consistently found greatest downwear to occur

at top of the platform where abrasion from beach material occurs (Ellis, 1986; Andrews, 2001; Foote et al., 2006; Moses and Robinson, 2011). Furthermore, a regional assessment of cliff retreat in California found that cliffs fronted by beaches retreated nearly 50% further than cliffs without beaches (Young, 2018). In contrast, beaches and fallen cliff debris can also act to dissipate wave energy that reduces platform downwear and wave impact at the cliff base, which slows cliff retreat (Sunamura, 1992; Walkden and Hall, 2005). These contrasting impacts on erosion as a result of beach material make it difficult to

understand their role in long-term rock coast development.

The coastal evolution model used here (Matsumoto et al., 2016) does not consider the impact of beach material on cliff retreat rates. The combination of across-shore and along-shore processes involved with transport of beach material and limited data on long-term beach evolution has meant that the role of beaches in the long-term evolution of rock coasts remains uncertain

and understudied (Naylor et al., 2014; Hurst et al., 2016). As a result, only a select number of modelling studies have investigated interactions between rock coast evolution and beach dynamics. A beach profile was incorporated into the Soft Cliff And Platform Erosion (SCAPE) model as exclusively a protective feature (Walkden and Hall, 2005). Beach protection of the upper intertidal platform, and resultant decreased cliff retreat is countered with the presence of beach material steepening



the rock shore platform, which makes the shore platform and cliff more vulnerable to erosion (Walkden and Hall, 2005). Both
the abrasive and protective roles of beaches were incorporated into the Limber and Murray (2011) model, in which cliff retreat
increases with active abrasion and cliff retreat decreases with either limited beach sediment that prevents abrasion, or too much
sediment that prevents wave erosion.

At our chalk sites, there is evidence of beach thinning across the Holocene (Dornbusch et al., 2006a, 2008). It has also been
suggested that beach thinning has contributed to the recent accelerations in cliff retreat rates due to diminished protection and
abrasive material (Dornbusch et al., 2008; Hurst et al., 2016). Thicker and wider beaches in the past are likely to have played
a protective role and dampened wave erosion and slowed cliff retreat rates at these south coast chalk sites. Although widely
variable spatially and temporally, Dornbusch et al. (2008) suggested that past average beach widths of ~37 m were unable to
protect the cliff from erosion and that average beach widths need to exceed ~70 m to entirely prevent wave erosion, including
from storm waves, at the cliff. With observed beach widths <37 m and >70 m at local-scale, protection and abrasion must be
varied spatially across the south coast chalk cliffs. Due to our limited current knowledge, we are unable to quantify at what
time, on average, flint beaches along south coast chalk cliffs crossed the threshold from a protective feature to an abrasive one.
Our work highlights the importance of beach material at rock coasts sites and that we need to better understand feedbacks
between beach dynamics, wave erosion and cliff failure.


Although not modelled here, incorporation of beach material into the rock coast system will consistently lower the modelled
$^{10}$Be concentrations. The presence of beach material lowers $^{10}$Be concentrations by 1) shielding the platform from cosmic rays,
which prevents production in the shore platform (Regard et al., 2012; Hurst et al., 2017) and 2) abrading and removing the
surficial layers of rock with the highest $^{10}$Be concentrations to expose rock with lower concentrations beneath. Hurst *et al*.
(2017) found that beach widths must be >50 m to lower concentrations significantly (by >15%). Because beach cover has not
been incorporated within our dynamic model simulations, our model results may be overestimating long-term cliff retreat rates
because we model low concentrations to indicate fast cliff retreat rates without the influence of beaches (Regard et al., 2012;
Hurst et al., 2016, 2017). However, accounting for the influence of beaches would make long-term cliff retreat rates slower
than our estimations, and, in turn, the recent acceleration seen in historical cliff retreat rates even greater (Hurst et al., 2016).
Furthermore, because beach presence lowers CRN concentrations, the low $^{10}$Be concentrations measured nearest the beach at
Beachy Head and Seven Sisters could be caused by beach cover, recent cliff retreat accelerations, or a combination of the two.
At Beachy Head, the nearshore concentrations are ~39–74% lower than the best-fit model predictions (Fig. 4). Similarly, at
Seven Sisters, nearshore concentrations are ~57–84% lower than the best-fit model predictions. According to Hurst et al.
(2017), the beach width would need to exceed ~100 m for at least the past 600 years for model predictions to align with the
measured data at Beachy Head and Seven Sisters. Observed beach widths are 15–73 m at these four south coast sites, therefore,
even with thicker and wider beaches in the past (Dornbusch et al., 2006a, 2008), it is unlikely for overestimations in nearshore
concentrations to be exclusively as a result of shielding from beach material or cliff debris. Furthermore, even the most massive





cliff falls (>10,000$^3$) can transport material away in a number of decades (Mortimore et al., 2004b; Moses and Robinson, 2011).
Observed beach widths and transportation time of fallen cliff debris further supports the scenario of a recent acceleration in
cliff retreat rates, which may be responsible for the low $^{10}$Be concentrations that the dynamic model outputs are unable to fit
at Seven Sisters and Beachy Head. Nevertheless, beach thinning and resultant increased abrasion has the potential to enhance
cliff retreat rate and contribute to such an acceleration that would account for the low $^{10}$Be concentrations (Limber and Murray,
2011).

### 5.5.3 Shore platform stepped topography

At Hope Gap and, most notably, at Seven Sisters, the best-fit model results do not match the measured stepped increase in
elevation in the upper intertidal zone of the shore platform (Fig. 4). These steps found at the south coast chalk platforms are
caused by heterogeneous beds of chalk and flint that vary in thickness and material strength (Moses and Robinson, 2011).
However, the model's implementation of lithology assumes homogenous material resistance across the shore platform and
cliff with no consideration of stratigraphic layers (Matsumoto et al., 2016). Unlike at the previously studied sandstone sites
(Shadrick et al., 2021), observed local variations in lithology has greater control on the shore platform meso-morphology at
these chalk coasts sites, and this variation has not yet been incorporated into our dynamic model optimisation. Future work
should aim to include heterogeneous material resistance into the model to better replicate the topographic steps and associated
processes, such as step backwearing, at these chalk sites.


## 6 Conclusions

In this study, we have quantified transient, long-term cliff retreat rates across the late-Holocene for four chalk rock coast sites
in the south of England. We have achieved this through multi-objective optimisation of a process-based coastal evolution
model to measured topographic and $^{10}$Be CRN concentration data. An improved understanding of how cliff retreat rates
responded to past changes in RSL helps to inform models that forecast cliff retreat rate response to climate-change driven
accelerations in RSL rise.

We have compared millennial-scale cliff retreat rates quantified by a previous steady-state equilibrium coastal evolution model
(Hurst et al., 2016) to rates derived from our new transient, process-based coastal evolution model (Matsumoto et al., 2016)
for two sites on the Sussex coastline at Hope Gap and Beachy Head. Our results provide further support for previous findings
of a significant recent acceleration in cliff retreat rates compared to the long-term rates quantified for the late-Holocene at the
Sussex coastline (Hurst et al., (2016)). Measured historical rates of cliff retreat during the past ~130 years range from 22–32
cm yr$^{-1}$ for the three Sussex coast sites (Dornbusch et al., 2008). However, our model results optimised to $^{10}$Be concentration
and topographic data suggest cliff retreat rates during the past ~2000 years, were 6–18 cm yr$^{-1}$ across these three Sussex coast
sites. Long-term cliff retreat rates for these sites also track the rate of RSL across the Holocene, which is similar to results
found for two previously studied sandstone sites in the UK (Shadrick et al., in revision, 2021). However, the recent acceleration



in cliff retreat rates evidenced by historical observations suggest that the rate of RSL rise is not necessarily the greatest control on cliff retreat rates at our chalk rock coast sites. Optimised results from the process-based model suggest the observed cliff retreat rates quantified for the past ~150 years at the south coast chalk sites (Dornbusch et al., 2008) were last experienced

between 5300 and 6800 years ago when the rate of relative sea level rise was 5–8.7 times greater than the rate of RSL experienced during the observational record. Model results suggest that cliff retreat rates were as much as 110 cm yr$^{-1}$ when the rate of RSL rise was 2.6 mm yr$^{-1}$ at the Seven Sisters site 7000 years BP. For other chalk sites that we studied, including Hope Gap, Beachy Head and St Margaret's, when the rate of RSL was 2.6 mm yr$^{-1}$ 7000 years BP, cliff retreat rates ranged from 15 to 55 cm yr$^{-1}$.


It is important to understand the long-term processes acting at shore platforms because these landforms play a critical role in mediating cliff erosion. The application of a process-based model used here has identified contrasting results for relative intertidal weathering rates across shore platforms between relatively strong sandstone and relatively weak chalk rock types. The optimised model results suggest that the rate of intertidal weathering was ~2–3 orders of magnitude greater at the chalk

sites compared to the sandstone rock coast sites. However, at sandstone sites, optimised results suggest that negligible subaerial weathering had to occur to match the measured data. Whereas, at all chalk sites that we studied, active subaerial weathering was required to match the measured topography and [10]Be concentrations. Furthermore, our results found no significant differences between best-fit material resistances for the sandstone and chalk sites. This indicates that it is not the material strength of the lithology alone, but how weatherable the material is that is important in the long-term evolution of rock coast

sites.

Comparisons between modelled results and measured data also suggest heterogeneous lithologic resistance and beach presence play an important role in the control of shore platform morphology, resultant [10]Be concentrations, and as a result, the long-term evolution of chalk rock coasts. These results particularly illustrate the importance for future work to consider role of

beaches to ensure long-term cliff retreat rates are not overestimated from [10]Be concentrations.

Nevertheless, results provided here have advanced the understanding of the long-term drivers of rock coasts in different lithological settings, particularly intertidal weathering, and beach material. Using a process-based coastal evolution model to interpret [10]Be concentrations has allowed us to not only quantify long-term transient cliff retreat rates, but also to help inform

the 'wave versus weathering' debate across millennial timescales. This is one of the first applications of a process-based model used to interpret [10]Be CRN concentrations which has, in turn, identified contrasts in prevailing long-term erosional mechanisms at coasts with different lithologies. Our findings highlight strong potential in our methodology to quantify long-term drivers of rock coast erosion at a variety of real-world sites. Quantifying cliff retreat rates and key erosion mechanisms across millennial timescales is especially important for rock coasts vulnerable to climate change, such as the UK chalk south coast.




**Code and data availability.** Input datasets can be found in the Supplement. Input data files, plotting scripts, code and documentation can be found at https://doi.org/10.5281/zenodo.5645478 (Hurst et al., 2021).

**Author contributions.** J.R.S., M.D.H., M.D.P. and D.H.R. designed the research and analysed the data; J.R.S., D.H.R.,
M.D.H. and K.M.W. collected measured datasets; D.H.R., J.R.S., A.J.S., prepared samples in the laboratory; J.R.S. developed the optimisation routine with support from M.D.H and M.D.P.; K.M.W. performed AMS measurements; J.R.S., M.D.H., D.H.R. and M.D.P. wrote the paper.

**Competing interests.** The authors declare that they have no conflict of interest.

**Acknowledgements.** This work was financially supported by a studentship from the Natural Environmental Research Council (NERC) Science and Solutions for a Changing Planet Doctoral Training Partnership (DTP), with additional funding by the British Geological Survey (BGS) to J.R.S. We acknowledge the support from the Australian Government for the Centre for Acceleration Science at ANSTO through the National Collaborative Research Infrastructure Strategy (NCRIS) and Research Portal award 10955 to D.H.R.

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
