# Peer review of "Constraints on long-term cliff retreat and intertidal weathering at weak rock coasts using cosmogenic $^{10}\text{Be}$ , nearshore topography and numerical modelling"

_Earth Surface Dynamics, 2022_

## Author Comment (AC1)

Reviewer comments in *blue italics*, author responses in black text, revised text in green text within quotation marks

Summary of primary issues to be addressed in revision, if we are invited to do revise the manuscript:

1) Provide more detail about muon versus spallation production and inheritance correction (Reviewer 1)
2) Correct errors and reorganize tables for clarity (Reviewers 1 and 2, respectively)
3) Improve consistency of units and terms (Reviewer 1)
4) Improve link between results and discussion of beach cover (Reviewer 2)
5) Add a comparative discussion of chalk cliff retreat rates in Sussex and Normandy (Reviewer 2)
6) Fix minor typographical errors and add suggested references (Reviewers 1 and 2)
* * *
*RC1: 'Comment on esurf-2022-28', Anonymous Referee #1, 30 Jun 2022*

*1 General comments (following Esurf structure requirements).*

*Scientific significance: excellent.*

*The article comprises field measurements and numerical modelling of Earth surface processes, here the effect of waves and weathering on chalk cliffs (i.e. essentially interactions between the hydrosphere and the lithosphere). For these reasons, this research matches undoubtfully the scope of ESurf.*

*The substantial contribution of the article to scientific progress within the scope of ESurf is achieved by new methods (dynamic model of cliff retreats with optimized parameters based on observed $^{10}$Be concentrations) and by new data (long-term chalk rock cliff retreat rates data).*

*Scientific quality: excellent.*

*The historical v.s. long term retreat of chalk rock cliffs is here calculated, analyzed and compared with. appropriate approaches. The potentialities and limitations of the process-based modelling approach used to constrain long-term chalk cliff retreat are thoroughly discussed, as well as perspectives of improvement for the future.*

*Substantial conclusions are reached and are: 1) A confirmation of recent acceleration of cliff retreat in the south of England based on the investigated dataset, 2) The decoupling of the relation RSL rise – cliff retreat during the acceleration period of chalk cliff retreat, 3) the differential control of weathering processes on sandstone vs chalk rock cliff retreat, and 4) some advances in the "wave versus weathering" debate that tries to find the main controller of material removal from cliff over time.*

*It should be mentioned that this paper is not 100 % a stand-alone paper, and follows/re-use some methods and conclusions developed in Hurst et al., 2016, and mainly Shadrick et al. (2021), where a similar modelling approach was applied to rock coasts. A summary of the latter is provided in the present paper, but many references are made to it throughout the text. It is thus necessary to read Shadrick et al. (2021), to fully understand the whole paper.*

*The article is perhaps a bit unbalanced between the approaches for constraining historical cliff retreat and long-term cliff retreat, but this is fully justified as constraining the long-term cliff retreat demands a modelling exercise whose strengths, limitations and uncertainties are fully addressed.*

 ***Presentation quality: excellent.***

*Figures are well supporting the text and are easily understandable even at first sight. The text is written with a logical progression of ideas and with accessible but high-quality English language.*

***Wider Comments***

*I find the discussion and conclusion a bit too long to read a perfectly balanced article, as the discussion represents 9.5 pages out of 31 pages of main text, so more than 30 % of the total article. The point on the cliff debris is very interesting to read, but is perhaps too long, as it was not mentioned in the objectives at the end of the intro or in the abstract that the influence of beach' pavements on erosion would be studied. It leaves me the sensation that you put a strong emphasis on a discussion point that was not expected when reading your abstract or your objectives.*

To address this reviewer comment, we will mention cliff debris in the objectives at the end of the introduction section and abstract.

Abstract: "Interpretation of results suggest that beaches, cliff debris, and heterogenous lithology play an important but poorly understood role in the long-term evolution of these chalk rock coast sites."

End of introduction section: "Here, we compare results between different models and discuss the influence of, e.g., beach material and cliff debris, as well as cliff retreat rates derived from historical records, which helps to advance our understanding of how rock coasts evolved both in the long-term past and more recently.

*2 Major concerns*

*L309-311: What you are interested in is the point to point difference in CRN concentration, and therefore is the inheritance background to the biggest issue. But as you mention the way you treat inheritance, could you justify the reasonability of your assumption that the last exhumated sample contains almost only inherited $^{10}Be$ atoms? If your sample remains buried under the cliff for a substantially long time, then only Muon produced CRNs accumulate at an extremely low production rate, until a possible secular equilibrium, from where no net CRN accumulation occurs. How would this inherited CRN quantity actually compare with the ones produced via the surface production rate in your sample once it is exhumed?*

The author is correct that the inheritance of $^{10}Be$ (I.e. that there is some concentration already in the rock before it is exposed at the surface) is an important consideration when dealing with shore platform samples which have relatively low concentrations. Hurst et al. (2016) produced a figure to show the expected concentrations due to muon production as a function of depth below the cliff top for unexposed rock (see below). The two measured values from that paper are also plotted.

[Figure]

Figure 5 from Hurst et al. (2016): [10]Be concentrations with depth produced by deep-penetrating muons. Surface lowering rates range from 0-0.1 mm yr[-1]. Red circles show measured [10]Be concentrations, against the local cliff height (depth) for each site. Measured inheritance is consistent with surface lowering rates of 0.01-0.04 mm yr[-1].

Cliff height at Seven Sisters is 47m and thus we'd expect inherited [10]Be concentrations to be on the order of 2000-3000 atoms g[-1]. Sample SS09 has a concentration of ~2700 atoms [g-1]. Thus, in the absence of a shielded sample from the cliff base or within a cave, the nearest sample to the cliff must suffice.

*Moreover, in your results (L360-365), you get an order of magnitude for the Cliff retreat of, say, 10 cm/yr. Your sample SS09 that is used for inheritance background is located 41 m away from the current Cliff position (Table 2). Correct me if I am wrong, but in such circumstances you would get a CRN accumulation at the surface of your sample for (41 m/ 10 cm/yr =) 410 yrs. Should your surface production ratio be around 4-5 atoms/g/yr,, you would end up with 1500-2000 at/g produced after the exhumation of your sample, that are finally not attributable to inheritance. This is a substantial quantity if you compare it to the measured CRN concentration in your sample SS09 (2770 atoms/g). The partial shielding for water probably decreases the CRN production rate, but to what extent?*

*If this example would be exact, then your CRN data points on Figure 5 for the Seven Sisters' Panel should be diverted 1500-2000 at/g above their current value. From the naked eye, you would get*

*4 more observed data into your shaded area, and likely a better fit between simulated and observed CRN concentrations.*

*This is a point that may deserve some attention in the discussion.*

The reviewer highlights that the uncertainty in our inheritance correction could be substantial, though we would argue they overestimate the expected concentration accumulated after the cliff unveils the rock. Historical cliff retreat rates are 40 cm yr$^{-1}$ averaged over the last 150 years, so this is just 100 years of exposure (the recent acceleration in retreat rates has helped us in this regard. Production of $^{10}$Be is dampened on the upper shoreface due to water shielding as acknowledged, having an average scaling factor of ~0.75 (Hurst et al., 2017), and there is also the potential effects of beach/debris cover. Thus, we would expect at most, a difference of 300 atoms g$^{-1}$ between this sample and a true inheritance value. We will add more clarification about the Be-10 concentration of SS09 to strengthen the justification for using it as an accurate correction. Specifically, we will compare the concentration to the predicted Be-10 concentration in a subsurface sample at a depth equivalent to the observe cliff height to show that the concentration is consistent with the prediction (e.g., Fig 3C of Hurst et al., 2016).

We have added to the manuscript:

*"The concentration at SS09 is similar to that which would be expected directly beneath a 47m high cliff for a secular equillibrium between muon production at depth and radioactive decay. However, some minor (300-400 atoms g-1) additional 10Be accumulation may have occurred since the cliff retreat from the site of SS09 over the last 100 years."*

*L633-635: again (see comment on L309-311), if there is any overestimation of the observed inherited $^{10}$Be concentration (From Hurst et al., 2016, when I look at Fig. 3, I see that the closest CRN observed conc. is more than 50 m from the current cliff position), then you would have more of the $^{10}$Be conc. attributable to post-exhumation production, your $^{10}$Be conc in Fig. 5 would increase and thereby better match your shaded zone for the Beachy Head site. Is it a possibility?*

With this and the previous couple of comments, the review has articulated a key challenge of our work: that unteasing the surface production from the inherited longer-term muogenic production is not always straightforward. Similar to the above, we will add additional clarification and justification for the muon-produced Be-10 and its use as an inheritance correction.

**3 Specific comments**

*L82 : perhaps give cite the studies you are talking about.*

We will add Regard et al., 2012 and Hurst et al., 2016 references here.

"Previous CRN analysis at chalk rock coasts have given novel insights into the long-term cliff retreat, yet these studies have always assumed a simplistic, steady-state geometric model of coastal evolution (Regard et al., 2012; Hurst et al., 2016)."

*L187-189: Just a detail but i needed to read the paragraph L187-189 several times to make a clear difference between your 2 study sites + the two you reprocessed from Hurst et al. (2016) vs their associated GIA reference site. I think "the three sites on the Sussex coast, including Seven Sisters, Hope Gap and Beachy Head" is an ambiguous fomrulation as it leaves the impression that named sites are included in other sites. Perhaps rephrase slightly ?*

We will rephrase for clarity.

"RSL histories have been supplied from a glacial-isostatic adjustment (GIA) model (Bradley et al., 2011). The three sites, including Seven Sisters, Hope Gap and Beachy Head, all on the Sussex coast have the same reference location for the GIA model ('Sussex', (Bradley et al., 2011)). The GIA reference location for the St Margaret's site was 'Kent' (Bradley et al., 2011)."

*L202-203: First time that "multi-objective optimisation" appears in the text. The sentence is formulated as if this concept has been described or at least mentionned above in the text. Perhaps around L105 would be a right spot to refer to it a first time?*

We will add a reference to "multi-objective optimisation" around L105 and explain what we mean by this by referencing our previous publication and directing the reader to the appropriate section in the methods that gives a summary of the approach.

"This study expands on the application of CRN exposure analysis of shore platforms in the UK by investigating the long-term rate and nature of cliff retreat using our coastal evolution model and multi-objective optimisation approach (Shadrick et al., 2021; see section 3.1) for two new UK, chalk rock coast sites: Seven Sisters and St Margaret's."

*L228-229: Could you precise how you account for resp. spallogenic and muogenic contribtuion to $^{10}$Be production ? Currently, it sounds a bit like a 1D model with only 2 levels : the surface level where CRN production is only controlled by spallation (very reasonable approx.) and a depth*

*level where CRN production is controlled only by muones (reasonability of approximation depends on what depth we talk about? Is is it a few meters or tens of meters? This would probably change the assumptions). Could you clarify this ? Or alternatively, show the formulas you have used ?*

Production by spallation and muons both takes place at the surface and at depth and is governed by exponential equations with different depth dependencies. We only use a single exponential for muons, grouping the production due to fast and slow muons respectively (following Braucher et al 2013). We will add clarification about how the model accounts for spallogenic and muogenic Be-10 production.

We now state:

"Both spallation-produced 10Be at the surface and muon-produced 10Be are calculated as a function of depth are modelled by summing the exponential functions specific to spallation and muon production (Braucher et al., 2013)."

*And:*

*"A full description of the 10Be production model can be found in Hurst et al., (2017)."*

*L233: You refer here to the CRN concentration analyzed at the surface across the progressively exhumed platform I guess? Perhaps just mention it to avoid people getting confused between your horizontal profile and the depth profile you referred to on L229.*

We will add clarification here that we are referring to the horizontal profile.

"These factors combine to result in the predicted 'humped' 10Be concentration horizontal profile offshore."

*L235 : Same comment as for L228-229 : Which production pathways' distribution governs your simulated CRN depth profile?*

We will add clarification about how the model accounts for spallogenic and muogenic Be-10 production as per the previous comment for L228.

*See response to previous comment. Readers also pointed to Hurst et al. (2017) for a fuller description of the approach to calculating 10Be production.*

*L335: Table 2: If i get it well, inheritance was subtracted to the background corrected ¹⁰Be conc. (e.g., 5.06 - 2.77 = 2.29 for SS01)? Then some column calculation are not ok (e.g., SS05 is higher*

*after the correction for inhertiance which is not possible - i think the true value is rather 0.922 and not 9.22). Same for SS07 and SS08. Please check carefully each column.*

The reviewer is correct and we thank them for spotting this error. We will correct the values in the table accordingly.

| Sample ID | Location (British Nat. Grid) | | Distance from cliff (m) | Background-corrected concentration $^{10}$Be *(x $10^3$ atoms g$^{-1}$) | ± 1σ AMS Analytical uncertainty (x $10^3$ atoms g$^{-1}$) | Inheritance corrected $^{10}$Be ** (x $10^3$ atoms g$^{-1}$) | ± 1σ *** (x $10^3$ atoms g$^{-1}$) |
|---|---|---|---|---|---|---|---|
| | Easting (m) | Northing (m) | | | | | |
| SS01 | 553791 | 96526 | 219 | 5.06 | 0.23 | 2.29 | 0.28 |
| SS02 | 553813 | 96536 | 203 | 4.41 | 0.23 | 1.64 | 0.29 |
| SS03 | 553818 | 96555 | 185 | 4.88 | 0.25 | 2.10 | 0.30 |
| SS04 | 553827 | 96578 | 160 | 4.38 | 0.24 | 1.60 | 0.29 |
| SS05 | 553814 | 96598 | 146 | 3.70 | 0.20 | 0.92 | 0.26 |
| SS06 | 553812 | 96618 | 128 | 5.70 | 0.26 | 2.93 | 0.31 |
| SS07 | 553811 | 96639 | 110 | 3.11 | 0.17 | 0.33 | 0.24 |
| SS08 | 553824 | 96666 | 78 | 3.11 | 0.21 | 0.33 | 0.27 |
| SS09 | 553818 | 96708 | 41 | 2.77 | 0.15 | - | - |

**Table 2: $^{10}$Be sample and concentration data for Seven Sisters.**

*L350: Table 3: Do you use SM05 to correct for the inheritance ? you have a value of 1.97 x $10^3$ at/g but in the column where inheritance is subtracted, it seems only 1.29 x $10^3$ atoms/g were subtracted. Is there a reason for such a difference?*

We thank the reviewer for spotting this typo. We will correct the value for SM05 in the table to 1.29 +/- 0.28 x 10^3 at/g.

**Table 3: $^{10}$Be sample and concentration data for St Margaret's.**

| Sample ID | Location (British Nat. Grid) | Background-corrected | ± 1σ AMS Analytical | Inheritance corrected $^{10}$Be |
|---|---|---|---|---|

| | Easting (m) | Northing (m) | Distance from cliff (m) | Concentration $^{10}$Be *(x$10^3$ atoms g$^{-1}$) | uncertainty (x $10^3$ atoms g$^{-1}$) | ** (x $10^3$ atoms g$^{-1}$) | ± 1σ *** (x $10^3$ atoms g$^{-1}$) |
|---|---|---|---|---|---|---|---|
| SM05 | 636761 | 144089 | 0 | 1.29 | 0.28 | - | - |
| SM06 | 636779 | 143992 | 73 | 4.02 | 0.79 | 2.73 | 0.84 |
| SM07 | 636782 | 143991 | 77 | 4.71 | 0.81 | 3.41 | 0.86 |
| SM08 | 636789 | 143983 | 85 | 5.84 | 0.81 | 4.55 | 0.86 |
| SM09 | 636802 | 143974 | 99 | 6.26 | 0.82 | 4.97 | 0.87 |
| SM10 | 636812 | 143972 | 110 | 6.53 | 0.83 | 5.24 | 0.88 |
| SM04 | 636819 | 143973 | 116 | 7.23 | 0.84 | 5.93 | 0.89 |
| SM11 | 636823 | 143971 | 120 | 5.44 | 0.82 | 4.15 | 0.87 |
| SM03 | 636828 | 143968 | 126 | 5.62 | 0.83 | 4.33 | 0.88 |
| SM12-1 | 636835 | 143963 | 135 | 4.51 | 0.81 | 3.22 | 0.85 |
| SM12-2 | 636835 | 143963 | 135 | 5.84 | 0.81 | 4.54 | 0.86 |
| SM13 | 636845 | 143965 | 146 | 3.81 | 0.79 | 2.51 | 0.84 |
| SM01 | 636869 | 144009 | 153 | 10.94 | 0.91 | 9.65 | 0.94 |
| SM14 | 636859 | 143952 | 160 | 7.01 | 0.82 | 5.72 | 0.86 |
| SM02 | 636866 | 143953 | 167 | 7.99 | 0.88 | 6.71 | 0.92 |
| SM15 | 636887 | 143949 | 188 | 6.79 | 0.81 | 5.50 | 0.86 |

*L417-431: Perhaps try to be consistent with the units you use throughout the paper. In Table 2/Table 3, your CRN conc. is written scientifically (x 10³ atoms/g), whereas its is in atoms/g in the paragraph that starts on L417, and in k atoms/g in Fig. 5. Same with yrs BP, that are sometimes written in K yrs BP (Fig. 6, L460; Fig. 7, L571).  Could you homogenize the way units are written?*

We will make the units in the paragraph that starts L417 "x $10^3$ atoms gram$^{-1}$" to be consistent with thetables and Figure 5.

"In fact, the peak in modelled $^{10}$Be concentrations at Beachy Head is ~4.5 x $10^3$ atoms g$^{-1}$ lower than the measured $^{10}$Be concentration peak. The model also underestimates the peak in $^{10}$Be concentrations by ~2.5 x $10^3$ atoms g$^{-1}$ at St Margaret's. Moreover, at St Margaret's, a localised drop in measured $^{10}$Be concentrations ~120–150 m from the cliff

base is not captured in the modelled [10]Be concentrations. Similar to the topographic results, the [10]Be concentrations closest to the cliff base cannot be well matched, especially at the Beachy Head and Seven Sisters sites. Specifically, the model is overestimating the [10]Be concentrations by at most ~1 x 10$^3$ atoms g$^{-1}$ at ~50–150 m offshore from the cliff, in the upper-intertidal zone at both Beachy Head and Seven Sisters sites."

*L539: Table 6: it is a detail but in some cells you cite the rate values in a decreasing way (e.g. "7 to 3") and in some other in an increasing way (2.6 to 30.4). Is it because some rates tend to decrease with time, while some others increase with time? Is there a specific reason for this? If not, it would be better to cite everything in an increasing way. Also consider to put the same number of significant digits everywhere.*

The reviewer correctly identifies that the decreasing versus increasing values that we cite reflect deceleration versus acceleration; therefore, we retained the decreasing or increasing pattern in the values. However, we will address the reviewer' other comment by using the same number of significant figures throughout the table.

**Table 6: Comparisons between historical cliff retreat rates and long-term cliff retreat rates derived from both a dynamic model and a geometric model.**

**Note the decline in long-term transient cliff retreat rates follows the pattern of RSL rise rate and the long-term steady-state cliff retreat rates are a step change (see Fig. 7).**

| Site | Historical cliff retreat rates | | Long-term, dynamic cliff retreat rates | | Long-term, geometric cliff retreat rates | |
|---|---|---|---|---|---|---|
| | Time period (years BP) | Rate (cm yr$^{-1}$) | Time period (years BP) | Rate (cm yr$^{-1}$) | Time period (years BP) | Rate (cm yr$^{-1}$) |
| St Margaret's | 122 | 7 ± 4 | 5300 | 7 to 3 | - | - |
| Seven Sisters | 128 | 39 ± 4 | 1800 | 18 to 13 | - | - |
| Hope Gap | 128 | 32 ± 4 | 4100 | 10 to 3 | 4315 | 6 to 1 |
| Beachy Head | 128 | 22 ± 4 | 4000 | 10 to 4 | 6139 | 3 to 30 |

*L504-505: when you say "an order of magnitude increase" of your short-term retreat rate compared to your long-term one, I agree with it if you take the lower values for Beachy Head (22 v.s. 2.6), but not the highest one (22 v.s. 30.4). Perhaps give some nuance to your statement.*

We will revise this statement to be more nuanced and clearer; specifically, we will clarify that it is up to an order of magnitude increase at Beachy Head and that these data are from previous work (i.e., Hurst et al., 2016).

"Using the steady-state model, previous work by Hurst et al. (2016) identified up to an order of magnitude increase in recent cliff retreat rates compared to long-term, Holocene-average rates at Beachy Head and Hope Gap."

*L581: in the title of subsection 5.4, you announce that you are going to talk about erosion processes, but the following paragraphs treat about weathering. At some places in your text, e.g. L87 or L137-138, you use formulation like "weathering and erosion". In your model, a free parameter is the "maximum intertidal weathering rate" and in L217, you use the formulation weathering-driven erosion. In L50-55, you clearly state the difference between weathering and wave erosion, thereby implying that any physical erosion performed by another agent than waves is encompassed in the notion of weathering. This makes the first reading a bit confusing as i always needed to check when you talk about erosion if it is a contraction of "wave erosion" or if it is physical erosion "belonging" to weathering.*

*In many (soil) studies, weathering is only used for chemical processes, by opposition to physical erosion. The sum of (chemical) weathering and physical erosion corresponds to the denudation.*

*I am not sure how those notions are transposed in coastal geomorphology, but I think you would gain in being fully systematic with the term you use.*

Section 5.4 the word 'erosion' as been replaced by the more appropriate term 'weathering' for this section.

Our modelling does not distinguish specific weathering processes, all is lumped into a single weathering shape function, informed by the wetting and drying experiments of Porter et al (2010). The section in lines 50-55 highlighted by the reviewer we think do a good job of framing how we are thinking about these often-confused suites of processes. Weathering is the processes of rock weakening; erosion is the removal of weakened mass.

We have checked all instances of the words erosion and weathering to make sure we are talking about one or other or both.

When talking about both we have been more careful with language:

"Because flints are composed of diagenetic silica and are, therefore, more resistant to weathering (and by proxy, erosion) than the carbonate chalk itself"

"Intertidal weathering can be an important precursor for erosion, weakening the substrate so that less energy is required to mobilise rock mass."

**4 Technical corrections**

*L53: This sentence seems to want to oppose chemical weathering and physical erosion. Shouldn't the "and" from L53 be replaced by a "," ?*

We will make the suggested change.

"Where weathering relates to the weakening of the rock material through a combination of physical, chemical and biological processes, wave erosion relates to the physical removal of rock material from the shore platform surface and cliff by means of wave action."

*L89: typo: missing dot at the end of the paragraph*

We will make the suggested change.

"Furthermore, a rigorous optimisation routine is required to optimise a process-based coastal evolution model to high-precision $^{10}$Be CRN concentrations and topographic data (Shadrick et al., 2021)."

*L335: Table 2: A word seems to be missing in the header of your last column (1 sigma or so).*

We will make the suggested change.

| Sample ID | Location (British Nat. Grid) | | Distance from cliff (m) | Background-corrected concentration $^{10}$Be *(x $10^3$ atoms g$^{-1}$) | $\pm 1\sigma$ AMS Analytical uncertainty (x $10^3$ atoms g$^{-1}$) | Inheritance corrected $^{10}$Be ** (x $10^3$ atoms g$^{-1}$) | $\pm 1\sigma$ *** (x $10^3$ atoms g$^{-1}$) |
|---|---|---|---|---|---|---|---|
| | Easting (m) | Northing (m) | | | | | |
| SS01 | 553791 | 96526 | 219 | 5.06 | 0.23 | 2.29 | 0.28 |

| | | | | | | | |
|------|--------|-------|-----|------|------|------|------|
| SS02 | 553813 | 96536 | 203 | 4.41 | 0.23 | 1.64 | 0.29 |
| SS03 | 553818 | 96555 | 185 | 4.88 | 0.25 | 2.10 | 0.30 |
| SS04 | 553827 | 96578 | 160 | 4.38 | 0.24 | 1.60 | 0.29 |
| SS05 | 553814 | 96598 | 146 | 3.70 | 0.20 | 0.92 | 0.26 |
| SS06 | 553812 | 96618 | 128 | 5.70 | 0.26 | 2.93 | 0.31 |
| SS07 | 553811 | 96639 | 110 | 3.11 | 0.17 | 0.33 | 0.24 |
| SS08 | 553824 | 96666 | 78  | 3.11 | 0.21 | 0.33 | 0.27 |
| SS09 | 553818 | 96708 | 41  | 2.77 | 0.15 | -    | -    |

**Table 2: $^{10}$Be sample and concentration data for Seven Sisters.**

*L840: Robinson, D.a. => "a" should be upper case.*

We will make the suggested change.

Dornbusch, U. and Robinson, D. A.: Block removal and step backwearing as erosion processes on rock shore platforms: a preliminary case study of the chalk shore platforms of south-east England, Earth Surf. Process. Landf., 36, 661–671, https://doi.org/10.1002/esp.2086, 2011.

*Citation: https://doi.org/10.5194/esurf-2022-28-RC1*
* * *
*RC2: 'Comment on esurf-2022-28', Anne Duperret, 08 Sep 2022*

*General comments*

1. *Does the paper address relevant scientific questions within the scope of ESurf?*
2. *Does the paper present novel concepts, ideas, tools, or data?*
3. *Are substantial conclusions reached?*
4. *Are the scientific methods and assumptions valid and clearly outlined?*
5. *Are the results sufficient to support the interpretations and conclusions?*
6. *Is the description of experiments and calculations sufficiently complete and precise to allow their reproduction by fellow scientists (traceability of results)?*
7. *Do the authors give proper credit to related work and clearly indicate their own new/original contribution?*
8. *Does the title clearly reflect the contents of the paper?*
9. *Does the abstract provide a concise and complete summary?*
10. *Is the overall presentation well structured and clear?*
11. *Is the language fluent and precise?*
12. *Are mathematical formulae, symbols, abbreviations, and units correctly defined and used?*
13. *Should any parts of the paper (text, formulae, figures, tables) be clarified, reduced, combined, or eliminated?*
14. *Are the number and quality of references appropriate?*
15. *Is the amount and quality of supplementary material appropriate?*

1. *This paper is relevant to the scope of Esurf. Modelling of rocky coast erosion. Interaction between lithosphere, hydrosphere, atmosphere (weathering ).*
2. *Yes. New Data (new CRE data on two coastal chalk cliffs sites in Sussex and Kent, UK). The concept and modelling was already published by the same authors in Esurf in 2021. (Shadwick et al, 2021). It is usefull to read this previous paper to understand the purpose of this one. Acronyms are not detailed in this paper. This needs, such as MCMC, DSAS, GIA, RPM....*

To address this reviewer comment, we will make sure that these acronyms are defined where they are first used.

"Free parameters chosen to vary within the Markov chain Monte Carlo (MCMC) simulation were wave erodibility by means of wave height decay rate (y), material resistance (FR) and maximum intertidal weathering rate (K)."

"Here we used a similar approach to Dornbusch et al. (2008) and used the Digital Shoreline Analysis System (DSAS) 5.0 (Himmelstoss et al., 2018) to quantify cliff retreat rates between the years 1898 and 2020."

"RSL histories have been supplied from a glacial-isostatic adjustment (GIA) model (Bradley et al., 2011)."

3.   *Yes. Substantial conclusions are given.*
4.   *see 2.) Assumptions during the discussion are clearly and precisely described.*
5.   *yes. Results are sufficiant to support the interpretations and conclusions.*
6.   *Experiments and calculations used in the numeric modelling are explained in another paper (Shadrick et al, 2021)*
7.   *yes*
8.   *yes*
9.   *yes*
10. *yes*
11. *yes*
12. *see 2)*
13. *no*
14. *yes.*

*15.  yes. There is link to conduct to the code and data availibility and 2 tables S1 and S2 with brut RCE data on Seven Sisters and St Margaret's sites*

***Details comments***

*Line 30-195 : Introduction and background on south coast chalk cliffs is very well explained.*

*Line 80 : Change : Duguet, 2021 by Duguet et al, 2021*

We will make the suggested change.

"Using these techniques, both Hurst et al. (2016) and Duguet et al. (2021) identified modern accelerations in cliff retreat rates for chalk cliffs on the south coast of England and Normandy coast in France, respectively."

*Line 94 and in some other places of the text. Shadwick et al, in revision. Are you sure that this paper will be published ? Please, no citation in this paper because it is not possible to read it.*

This paper has now been published. We will update the reference accordingly.

"Long-term cliff retreat rates for the past 7000 years, and, in turn, projections of future cliff retreat rates to the year 2100 have been made for two sandstone, rock coast sites in the UK by combining the best available rock coast morphodynamic model with simulated CRN accumulation and optimising the model to high-precision CRN concentration measurements and topographic survey data (Shadrick et al., 2022)."

"Our results also reveal contrasting results between chalk and previously studied sandstone sites (Shadrick et al., 2022) regarding the key erosional mechanisms controlling long-term evolution."

*Line 137 : Add : Mortimore et al, 2001*

We will add the suggested reference.

"The Cretaceous white chalk cliffs and shore platforms contain bands of nodular and continuous sheets of flint parallel to bedding (Robinson, 2020; Mortimore et al. 2001)."

*Line 139 : I am not sure that some paleo-sand barriers are located offshore the Sussex coast.*

Apologies, this was a mistake, and now says "palaeo gravel barriers" as per the references cited.

*Line 285-290 : Cross-shore topographic profiles cover mainly the aerial part of the shore platform ? Elevation scales extend from zero (equivalent to cliff platform junction) to about –6 m (as shown on Fig. 5).*

We will clarify that the topographic profiles cover mainly the subaerial part of the shore platform exposed at low spring tides, but these have been combined with multibeam bathymetric surveys.

*"The Channel Coast Observatory (Channel Coastal Observatory, 2021) provides high quality coastal LiDAR and multibeam bathymetry that were combined, and and bathymetry imagery, and so we used this data to constructfrom which the an intertidal topographic cross-section 10 m wide swath profile was extracted to a distance of 300 m from the mdoern cliff."*

*Table 3 : A data presentation as a function of the cliff distance could be more easier to follow.*

We will revise the table to be ordered as a function of increasing distance from the cliff.

**Table 3: $^{10}$Be sample and concentration data for St Margaret's.**

| Sample ID | Location (British Nat. Grid) | | Distance from cliff (m) | Background-corrected Concentration $^{10}$Be *(x$10^3$ atoms g$^{-1}$) | ± 1σ AMS Analytical uncertainty (x $10^3$ atoms g$^{-1}$) | Inheritance corrected $^{10}$Be ** (x $10^3$ atoms g$^{-1}$) | ± 1σ *** (x $10^3$ atoms g$^{-1}$) |
|---|---|---|---|---|---|---|---|
| | Easting (m) | Northing (m) | | | | | |
| SM05 | 636761 | 144089 | 0 | 1.29 | 0.28 | - | - |
| SM06 | 636779 | 143992 | 73 | 4.02 | 0.79 | 2.73 | 0.84 |
| SM07 | 636782 | 143991 | 77 | 4.71 | 0.81 | 3.41 | 0.86 |
| SM08 | 636789 | 143983 | 85 | 5.84 | 0.81 | 4.55 | 0.86 |
| SM09 | 636802 | 143974 | 99 | 6.26 | 0.82 | 4.97 | 0.87 |
| SM10 | 636812 | 143972 | 110 | 6.53 | 0.83 | 5.24 | 0.88 |
| SM04 | 636819 | 143973 | 116 | 7.23 | 0.84 | 5.93 | 0.89 |
| SM11 | 636823 | 143971 | 120 | 5.44 | 0.82 | 4.15 | 0.87 |
| SM03 | 636828 | 143968 | 126 | 5.62 | 0.83 | 4.33 | 0.88 |
| SM12-1 | 636835 | 143963 | 135 | 4.51 | 0.81 | 3.22 | 0.85 |
| SM12-2 | 636835 | 143963 | 135 | 5.84 | 0.81 | 4.54 | 0.86 |
| SM13 | 636845 | 143965 | 146 | 3.81 | 0.79 | 2.51 | 0.84 |
| SM01 | 636869 | 144009 | 153 | 10.94 | 0.91 | 9.65 | 0.94 |
| SM14 | 636859 | 143952 | 160 | 7.01 | 0.82 | 5.72 | 0.86 |
| SM02 | 636866 | 143953 | 167 | 7.99 | 0.88 | 6.71 | 0.92 |
| SM15 | 636887 | 143949 | 188 | 6.79 | 0.81 | 5.50 | 0.86 |

*Line 402 : Change Fig. 4 by Fig 5*

We will make the suggested change.

Commented [MH1]: this is unclear to me, does the reviewer jsut want us to move fig 4 to be nearer to fig 5?

Commented [RH2R1]: The figure mistakenly referred to Fig 4, so the reviewer wanted it to be changed to the correct Fig 5. Done now.

"Comparisons between the modelled output and measured data is shown using the best-fit results and uncertainty defined by the 50%–50% weighted topographic–10Be concentrations MCMC results (Fig. 5)."

*Figure 5 : It appears that samples cover only the intertidal shore platform and they do not reach the main steps at the end of the shore platform except at Hope Gap. Unfortunately, 10 Be content do not seems to be perturbated around this step.*

*If a proposed explanation is given line 740, please make a link with the 10Be content.*

We will clarify that the samples cover only the intertidal shore platform due to sampling limitations. There is some evidence that steps in the topographic profiles correspond to changes in 10Be concentrations. We now add:

"At Hope Gap (see Fig. 5), there is a noteable step in the topography at a distance of ~180 m from the modern cliff. The next samples offshore from this location (200-250 m from the cliff) were sampled from lower elevations and have a lower concentration"

*How the authors explain the very localised high 10Be contents at St Margarets and Beachy Head ?*

*There is inherently some variability in measured concentrations, relative to the simplified representation in the model. These may be the result of limited surface lowering locally, reflecting heterogeneous, we add to the discussion:*

*"Such developments might also help to better match the relatively high concentrations at St. Margaret's and Beachy Head, that might reflect limited surface lowering locally"*

*Although it is also worth noting that we interpreted this signal at Beachy Head as a signal of a previously slower rate of cliff retreat in the past.*

We will provide a clearer explanation of the localized high Be-10 contents at these sites.

*Line 412 to 415 and Fig. 5 : Could the higher elevation of the shore platform topography versus the model uncertainty resulting from a thick shingle berm on the beach ? It is a typical accumulation at the top of the shore plateform, occulting cosmogenic signal.*

We will further address and clarify this point by directing the reader here to section 5.5.2, where this point is comprehensively discussed.

"These best fit results relative to the uncertainty range suggests the model was unable to move into a parameter space in the MCMC inversion to match these elevations while trying to simultaneously match the $^{10}$Be concentrations (see section 5.5.2 for discussion)."

*Line 440 to 450 : It is not so clear how long-term cliff retreat rate are calculated.*

*We now say:*

" The long-term trend in these new cliff retreat rates calculated from the optimised models across the past 7000 years reflect the trend in the rate of RSL for all four chalk sites"

*Line 567 to 570 : The Beachy Head inshore low concentrations could not be due to the debris of the large cliff collapse covering a large part of the inshore platform at this site, since 1999 at least?*

The reviewer is correct that shielding by sediment/beach deposits is a concern, and this was considered carefully in Hurst et al., (2016 and 2017). We will further address and clarify this point by directing the reader here to section 5.5.2, where this point is carefully discussed.

"Moreover, the best-fit $^{10}$Be concentration results found for the dynamic model at Beachy Head could not match the inshore low concentrations (Fig 5; see section 5.5.2 for discussion)."

**Conclusions**

*Line 760-774 : On the UK chalky coast, the comparisons between long-term erosion rates (between 5300 and 6800 years) and historical rates (about 150 years)*

*Models gives cliff retreat rates ranging between 15 to 55 cm/year at 7000 years BP at Hope Gap, Beachy Head and St Margaret's and 110 cm/year at Seven Sister's at 7000 years BP. Where comes from this discrepancy, taking into account the SLR was about the same on these sites at this period (2.6 mm/year) ?*

Simulations of RSL based on GIA modelling show rates of SLR that were much more rapid up to 7ka and have been declining since. Our modelling does not take account of any recent upturn in sea level rise (in recent decades) and thus by the end of the simulations SLR is < 0.5 mm/yr. Differences between the chalk sites may relate to different lithological properties (the chalk at St. Margeret's is more massive, while in East Sussex it is more

fractured) and possibly also relate to differences in hydrodynamic conditions, with East Sussex being more exposed.

*This is covered in the discussion by stating:*

*"The slower historical cliff retreat rates calculated at St Margaret's in comparison to the Sussex coast sites could be as a result of the harder Lewes Nodular chalk lithology at St Margaret's, in comparison to the soft, low-density, more densely fractured and high-porosity Seaford Chalk Formation on the Sussex coast (Mortimore et al., 2004a)."*

*When compared with historical rates calculated on about 130 years, cliff retreat rates vary between 22-32 cm/year. Are the authors conclude to a recent acceleration of the cliff retreat rates on Sussex chalky coast (at Hope Gap and Beachy Head) but not at Seven Sisters ?*

The conclusions now clearly state that the acceleration can be evidenced at all three sites:

*"Measured historical rates of cliff retreat during the past ~130 years range from 22–40 cm yr-1 for the three Sussex coast sites (Dornbusch et al., 2008). However, our model results optimised to 10Be concentration and topographic data suggest cliff retreat rates during the past ~2000 years, were 6–18 cm yr-1 across these three Sussex coast sites"*

*Line 776-785 : It is not surprising to conclude that weathering is more intense on chalk coast than on sandstone rock coast. Authors give an order of magnitude and it is great.*

*Finally, recent work conducted on chalk rock coast in Normandy, France (Duguet et al, 2021) conclude also to an acceleration of cliff retreat rates during the Holocene, using static model of erosion, based on a detailed submarine bathymetry analysis offshore the studied sites. A continuous submarine step is observed underwater and the 10Be content of some underwater samples has been analysed.*

*Some comparison of results could be discussed between chalk cliff retreat rates in Sussex and Normandy. It is roughly the same cliffs! A comparative discussion of results could be very interesting to complete our knowledge of chalk cliff retreat rates on each side of the Channel, through various periods of the past.*

*We have added a sentence to make this comparison and would welcome bilateral discussion on linking this up more in future.*

*"This is consistent with another recent study that has suggested recent acceleration in cliff retreat rates based on 10Be concentrations measured on chalk shore platforms (including sub-tidal measurements) on the French Coast of the English Channel at Mesnil-Val (Duguet et al., 2021)."*

*The question is : Do you think that a similar work in UK with underwater complementary bathymetry and CRE analysis, may help to better define erosive mechanisms and erosion rates versus SLR rates along chalky coast ? Pre-Holocene periods of inheritance needs also to be better defined along these coast-types, because even if the intertidal shore platform do not indicate re-occupation due to the low content of 10Be, it could be different off the shore.*

The reviewer makes a good point here, but it does not require any revision.

*I suggest minor revisions for this paper. It is an interesting peace of work. This needs, in my opinion, to be continued with some complementary work on these coasts, but it is another work! I suggest also to precise the conclusions, with less doubt.*

***Citation***: *https://doi.org/10.5194/esurf-2022-28-RC2*